# Learning Molecular Semantic Invariant Representation with Prototype Constraint

**Zhiqiang Li** [1]   **Jianqing Liang** [2]   **Zhiqiang Wang** [1]   **Xizhao Luo** [3]   **Jiye Liang** [1]

## Abstract

Molecular representation learning has achieved remarkable progress in molecular property prediction, yet out-of-distribution (OOD) generalization remains challenging. In practice, training data typically cover only a limited portion of the chemical space, causing models to rely on environment-dependent factors that fail to transfer when scaffold structures or functional compositions shift. To address this issue, we propose MoSIR, a framework for learning molecular semantic invariant representation with prototype constraint, which projects entangled molecular embeddings into a learnable semantic prototype space to extract semantic invariant representation while isolating environment-sensitive variations. Building upon this decomposition, we optimize a bi-level min-max objective that introduces representation perturbations to simulate plausible environment shifts and enforce semantic stability. We further provide theoretical guarantees for MoSIR by deriving an OOD generalization bound under distribution shifts. Extensive experiments on multiple molecular OOD benchmarks demonstrate that MoSIR consistently outperforms strong baselines across diverse shift settings, and qualitative analyses confirm that the learned prototypes capture meaningful chemical semantics.

## 1. Introduction

Molecular representation learning has become a cornerstone of computational chemistry and drug discovery, enabling effective prediction of molecular properties such as bioactivity, toxicity, and physicochemical attributes (Muratov et al., 2020; Macalino et al., 2015). Recent GNN-based (Kipf & Welling, 2017) molecular representations have achieved strong in-distribution performance on molecular property prediction tasks. However, despite these advances, the generalization of molecular representations under OOD scenarios remains a fundamental challenge (Yang et al., 2022).

In real-world applications, available training datasets cover only a small fraction of the vast chemical space. When test molecules exhibit systematic distribution shifts, such as changes in scaffold structures, substituent patterns, or functional group compositions, models trained under empirical risk minimization tend to rely on spurious, environment-dependent correlations, and therefore struggle to maintain stable performance under new data distributions (Fan et al., 2023). This issue is particularly pronounced in real-world drug discovery pipelines (Mendez et al., 2019), where molecular distributions may shift due to newly explored scaffolds, evolving therapeutic targets, or changes in experimental conditions(Wu et al., 2018; Muandet et al., 2013).

A growing body of work attempts to improve molecular OOD generalization through data augmentation or invariant representation learning. Contrastive pretraining methods construct multi-view molecular representations to enhance robustness to perturbations (Chen et al., 2024; Wang et al., 2022), while invariant learning approaches enforce representation consistency across environments or substructure partitions (Zhuang et al., 2023; Wu & Deng, 2025). Despite promising results under certain settings, many existing methods depend on predefined environment partitions or observed training environments, which limits their ability to capture unknown distributional variations in realistic chemical spaces. Moreover, directly enforcing invariance on entangled molecular representations can be brittle: the learned features may still mix stable chemical semantics with shift-sensitive structural variations, making invariance difficult to achieve under complex shifts.

In this work, we build on the observation that molecular properties are primarily governed by semantic factors such as functional group composition and high-level chemical motifs, while scaffold-level or geometric variations often

---

[1]Key Laboratory of Computational Intelligence and Chinese Information Processing of Ministry of Education, School of Computer and Information Technology, Shanxi University, Taiyuan, China [2]School of Computer Science and Engineering, Southeast University, Nanjing, China [3]School of Computer Science and Technology, Soochow University, Suzhou, China. Correspondence to: Jianqing Liang <liangjq@seu.edu.cn>.

*Proceedings of the 43rd International Conference on Machine Learning*, Seoul, South Korea. PMLR 306, 2026. Copyright 2026 by the author(s).

act as environment-specific nuisance factors. This motivates a representation learning principle. OOD generalization should be achieved by stabilizing molecular semantics rather than the entire entangled representation. To operationalize this idea, we propose MoSIR, a framework for learning molecular semantic invariant representations via a learnable prototype constraint.

MoSIR introduces a semantic prototype dictionary to project entangled molecular embeddings into an explicit semantic space, yielding an invariant semantic component and an environment-dependent residual. Importantly, we model distribution shifts by applying adversarial perturbations only in the residual space, which simulates plausible environment variations while preserving semantic anchoring. We then formulate learning as a bi-level min–max optimization, where the inner maximization finds residual perturbations that induce the largest semantic drift, and the outer minimization updates the model to maintain both predictive accuracy and semantic stability under such worst-case shifts. In addition, we derive an OOD generalization bound under distribution shifts and show that the prototype-based semantic bottleneck reduces hypothesis complexity, providing theoretical justification for the proposed design. Our contributions are:

- We propose a learnable prototype-based decomposition that separates molecular representations into an invariant semantic component and an environment-dependent residual, mitigating spurious correlations induced by distribution shifts.

- We formulate molecular OOD learning as a bi-level min–max optimization that actively simulates environment shifts by perturbing only the residual component, and explicitly enforces semantic invariance.

- We provide an OOD generalization bound highlighting the complexity reduction effect of prototype bottlenecking, and demonstrate consistent improvements over strong baselines on multiple molecular OOD benchmarks under diverse shift settings.

## 2. Related Work

**Molecular Representation Learning.** Molecular representation learning aims to encode molecules from different modalities, including SMILES strings, molecular fingerprints, and molecular graphs. Graph neural networks (GNNs), such as GCNs (Kipf & Welling, 2017), directly model molecular topology and node attributes, and the Message Passing Neural Network (MPNN) framework (Gilmer et al., 2017) further unifies this paradigm by iteratively aggregating local structural information to produce molecule-level representations. Recent self-supervised approaches leverage contrastive learning over augmented molecular

graphs, achieving improved sample efficiency and downstream performance (Wang et al., 2022). However, most existing methods still struggle to generalize under distribution shifts, since molecular representations often entangle stable chemical semantics with environment-dependent structural variations such as scaffold or functional composition changes. As a result, models may rely on spurious correlations that are predictive in training environments but fail to transfer to unseen test distributions.

**Molecular OOD Generalization.** OOD generalization is a key challenge in molecular learning due to distribution shifts across scaffolds, datasets, and experimental conditions. Benchmarks such as GOOD (Gui et al., 2022) and DrugOOD (Ji et al., 2023) show that GNNs with strong in-distribution performance often fail under scaffold or domain shifts. Invariant representation learning seeks features that are predictive and stable across environments (Chang et al., 2020; Creager et al., 2021), with connections to causal modeling (Mahajan et al., 2021; Lv et al., 2022; Wang et al., 2023) and representation disentanglement (Li et al., 2022; Zhang et al., 2022; Wu et al., 2022a). However, most existing methods target Euclidean data and do not readily extend to molecular graphs. In the molecular domain, many approaches adopt a "separate-then-encode" paradigm (Yang et al., 2022; Chen et al., 2022), which depends heavily on disentanglement quality and degrades for complex structures. iMoLD (Zhuang et al., 2023) follows an "encode-then-separate" strategy, but still struggles to extract invariant representations in challenging cases. CFD (Wu & Deng, 2025) leverages concept-level feedback to mitigate spurious correlations. Yet, its performance relies on high-quality concept signals and lacks active shift simulation.

## 3. Method

### 3.1. Problem Definition

We study OOD generalization for molecular representation learning. Let $\mathcal{G}$ denote the space of molecular graphs and $\mathcal{Y}$ the label space. Given a molecular graph $G \in \mathcal{G}$, the goal is to learn a predictor $f : \mathcal{G} \to \mathcal{Y}$ that accurately predicts its property label $Y \in \mathcal{Y}$. Molecular data are often collected under diverse experimental or chemical conditions, which can be abstracted as environments. Let $\mathcal{E}_{\text{all}}$ denote the set of all environments, and assume samples are drawn from environment-specific distributions $P(G, Y \mid E = e)$, where environment annotations are unavailable during training. Consequently, the training and testing distributions differ, i.e., $P_{\text{train}}(G, Y) \neq P_{\text{test}}(G, Y)$. The desired predictor should generalize across environments by minimizing the expected risk over a test-time environment mixture:

$$f^{\star} = \arg\min_{f} \; \mathbb{E}_{e \sim \pi(e)} \, \mathbb{E}_{(G,Y) \sim P(\cdot | E=e)} \big[ \ell(f(G), Y) \big],$$
(1)

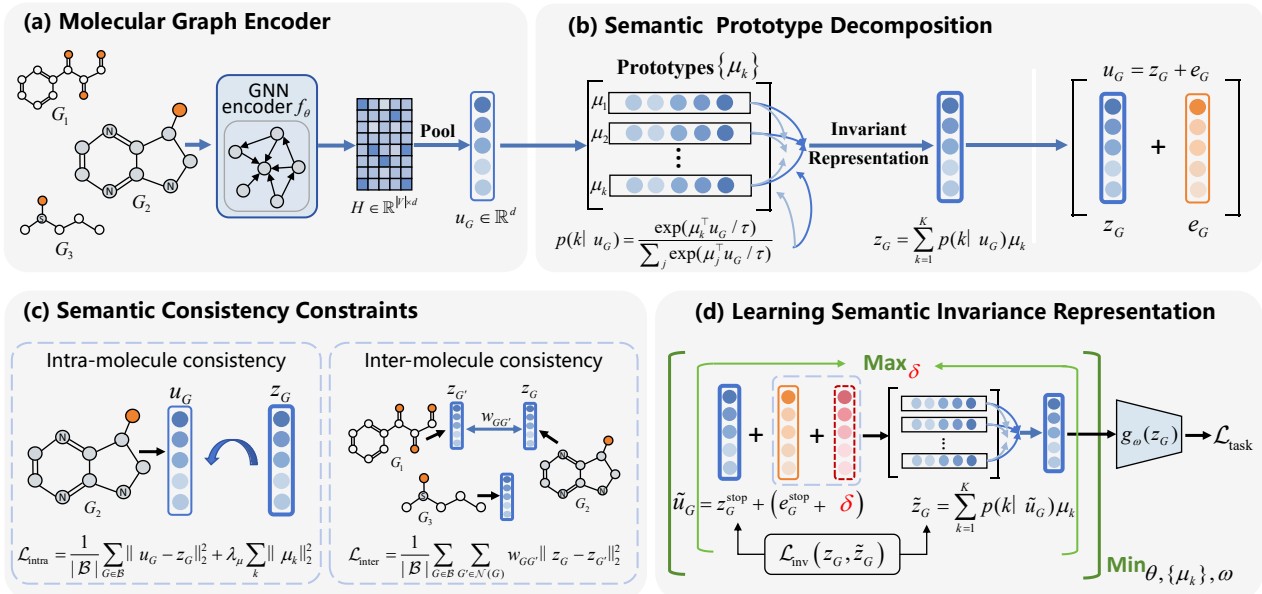

*Figure 1.* Overview of the proposed MoSIR framework. (a) A GNN encodes each molecular graph into an entangled representation $u_G$. (b) $u_G$ is projected onto a learnable semantic prototype dictionary to obtain the semantic invariant representation $z_G$ and the environment-dependent residual $e_G$. (c) Semantic consistency constraints regularize the prototype space by enforcing intra-molecule reconstruction and inter-molecule semantic smoothness. (d) Representation perturbations on $e_G$ simulate environment shifts, and a bi-level min–max objective enforces the invariance of $z_G$ while supporting property prediction.

where $\pi(e)$ denotes the environment mixture at test time and $\ell(\cdot, \cdot)$ is the task loss. By factorizing $P(G, Y) = P(Y \mid G)P(G)$, distribution shifts can be categorized into two types. Covariate shift corresponds to changes in molecular structure distributions while keeping the labeling mechanism invariant:

$$P_{\text{train}}(G) \neq P_{\text{test}}(G), \quad P_{\text{train}}(Y \mid G) = P_{\text{test}}(Y \mid G), \tag{2}$$

whereas concept shift occurs when the structure–property relationship varies across environments:

$$P_{\text{train}}(Y \mid G) \neq P_{\text{test}}(Y \mid G), \quad P_{\text{train}}(G) = P_{\text{test}}(G). \tag{3}$$

We evaluate both types of shifts in our experiments, and Figure 1 provides an overview of the proposed MoSIR framework.

### 3.2. Molecular Graph Encoding

Given a dataset $\mathcal{D} = \{(G^{(n)}, Y^{(n)})\}_{n=1}^{N}$, each molecule is represented as a graph $G = (V, E, X)$, where $V$ denotes atoms, $E$ denotes chemical bonds, and $X \in \mathbb{R}^{|V| \times d_x}$ denotes node features such as atom types, valence, and aromaticity. The target $Y$ corresponds to either a classification or regression property.

We first employ a GNN encoder $f_\theta$ to compute node representations via message passing:

$$H = f_\theta(G), \quad H = \{h_i \in \mathbb{R}^d \mid i \in V\}, \tag{4}$$

where $d$ is the hidden dimension and $\theta$ denotes model parameters. Each node embedding aggregates information from its local chemical neighborhood, capturing both property-relevant semantics and environment-dependent structural patterns.

We then apply a permutation-invariant readout function to obtain a graph-level embedding:

$$u_G = \text{Pool}(H) \in \mathbb{R}^d, \tag{5}$$

where $\text{Pool}(\cdot)$ can be instantiated as mean pooling. We refer to $u_G$ as an entangled representation, since it does not explicitly disentangle stable chemical semantics from shift-sensitive nuisance factors, which may lead to spurious correlations under distribution shifts.

### 3.3. Semantic Prototype Decomposition

To explicitly capture shared chemical semantics across molecules, we introduce a set of $K$ learnable semantic prototypes:

$$\mathcal{M} = \{\mu_k \in \mathbb{R}^d\}_{k=1}^{K}, \tag{6}$$

where each prototype $\mu_k$ represents a latent semantic concept shared across molecules such as functional-group-like motifs or reactivity cues. Unlike handcrafted fragments, these prototypes are learned end-to-end, enabling flexible discovery of property-relevant semantics.

Given the entangled embedding $u_G$, we compute a soft

assignment over prototypes:

$$p(k \mid u_G) = \frac{\exp(\mu_k^\top u_G/\tau)}{\sum_{j=1}^K \exp(\mu_j^\top u_G/\tau)}, \qquad (7)$$

where $\tau > 0$ is a temperature parameter controlling assignment sharpness. The distribution $p(k \mid u_G)$ can be interpreted as a semantic activation profile describing how the molecule is composed of prototype semantics.

We define the semantic invariant representation as the prototype mixture:

$$z_G = \sum_{k=1}^K p(k \mid u_G)\mu_k \in \mathbb{R}^d. \qquad (8)$$

This projection maps $u_G$ into a shared semantic coordinate system spanned by $\{\mu_k\}$. Importantly, $z_G$ serves as the **only representation used for prediction and invariance learning**, acting as a semantic bottleneck that encourages the model to rely on stable chemical semantics.

### 3.4. Semantic Consistency Constraints

While prototype projection provides a semantic bottleneck, the prototype space may still admit degenerate solutions without proper regularization. We therefore introduce semantic consistency constraints to stabilize learning and encourage meaningful semantic structure.

**Intra-molecular semantic consistency.** We encourage $z_G$ to preserve the dominant information in $u_G$ by minimizing reconstruction discrepancy:

$$\mathcal{L}_{\text{intra}} = \frac{1}{|\mathcal{B}|} \sum_{G \in \mathcal{B}} \|u_G - z_G\|_2^2 + \lambda_\mu \sum_k \|\mu_k\|_2^2, \qquad (9)$$

where $\mathcal{B}$ denotes a mini-batch. The first term ensures that the semantic projection explains the entangled embedding, preventing $z_G$ from becoming uninformative. The second term regularizes prototype magnitudes to improve numerical stability. We define the residual component as:

$$e_G = u_G - z_G. \qquad (10)$$

Since $\|u_G - z_G\|_2^2 = \|e_G\|_2^2$, minimizing $\mathcal{L}_{\text{intra}}$ encourages the model to concentrate property-relevant information into the semantic space $z_G$, rather than encoding it in the residual $e_G$.

**Inter-molecular semantic consistency.** OOD generalization further requires that semantically similar molecules form coherent neighborhoods in the invariant space. We construct a semantic neighborhood within each mini-batch and define the normalized neighborhood weight:

$$w_{GG'} = \frac{\exp(\cos(z_G, z_{G'})/\rho)}{\sum_{H \in \mathcal{N}(G)} \exp(\cos(z_G, z_H)/\rho)}, \qquad (11)$$

where $\cos(z_G, z_{G'}) = \frac{z_G^\top z_{G'}}{\|z_G\|_2 \|z_{G'}\|_2}$ and $\rho > 0$ controls distribution sharpness. The inter-molecule semantic smoothness loss is:

$$\mathcal{L}_{\text{inter}} = \frac{1}{|\mathcal{B}|} \sum_{G \in \mathcal{B}} \sum_{G' \in \mathcal{N}(G)} w_{GG'} \|z_G - z_{G'}\|_2^2. \qquad (12)$$

This objective encourages molecules with similar semantic compositions to stay close in the prototype space, forming a structured and transferable semantic manifold. We combine the two terms as:

$$\mathcal{L}_{\text{sem}} = \lambda_{\text{intra}} \mathcal{L}_{\text{intra}} + \lambda_{\text{inter}} \mathcal{L}_{\text{inter}}. \qquad (13)$$

### 3.5. Learning Semantic Invariant Representation

In molecular OOD settings, distribution shifts are mainly induced by systematic changes in environment-dependent structures such as scaffold variations. We therefore model unseen environments by introducing adversarial perturbations in representation space, and enforce semantic invariance against such environment shifts.

We decompose $u_G = z_G + e_G$ and apply adversarial perturbations only to the residual component:

$$\tilde{u}_G = \text{sg}(z_G) + \big(\text{sg}(e_G) + \delta\big), \quad \|\delta\|_2 \le \epsilon, \qquad (14)$$

where $\delta$ denotes an environment perturbation and $\text{sg}(\cdot)$ is the stop-gradient operator. This design prevents the adversary from directly altering the semantic anchor $z_G$, while allowing it to explore environment-sensitive variations through $e_G$. As a result, the perturbation simulates plausible environment shifts without collapsing the semantic coordinate system.

We recompute the semantic representation from the perturbed embedding: $\tilde{z}_G = \sum_{k=1}^K p(k \mid \tilde{u}_G)\mu_k$, which reflects the semantic drift induced by an adverse environment shift.

We enforce that the semantic representation remains stable under perturbations by treating $(z_G, \tilde{z}_G)$ as a positive pair and using other molecules in the batch as negatives:

$$\mathcal{L}_{\text{inv}} = \frac{1}{|\mathcal{B}|} \sum_{G \in \mathcal{B}} -\log \frac{\exp(\cos(z_G, \tilde{z}_G)/\gamma)}{\sum_{G' \in \mathcal{B}} \exp(\cos(z_G, \tilde{z}_{G'})/\gamma)}, \qquad (15)$$

where $\gamma > 0$ is a temperature. This objective explicitly penalizes semantic drift under environment perturbations, encouraging stable and transferable molecular semantics.

Finally, we learn semantic invariant representations by minimizing task loss and semantic regularization under worst-case residual perturbations:

$$\min_{\theta, \{\mu_k\}, \omega} \mathbb{E}_{\mathcal{B} \sim \mathcal{D}} \Big[ \mathcal{L}_{\text{task}} + \mathcal{L}_{\text{sem}} + \lambda_{\text{inv}} \max_{\|\delta\|_2 \le \epsilon} \mathcal{L}_{\text{inv}} \Big]. \qquad (16)$$

The inner maximization searches for perturbations that maximally disrupt semantic invariance, while the outer minimization updates the model to maintain predictive performance and semantic stability under such worst-case shifts.

In practice, the inner maximization is approximated by projected gradient ascent:

$$\delta^{(t+1)} = \Pi_{\|\delta\|_2 \leq \epsilon}\left(\delta^{(t)} + \eta_\delta \nabla_\delta \mathcal{L}_{\text{inv}}\right), \qquad (17)$$

where $\Pi$ denotes projection onto the $\ell_2$-ball. For downstream molecular prediction, we use the learned semantic invariant representation $z_G$ as input to a prediction head $g_\omega$, with cross-entropy loss for classification and mean squared error for regression. Algorithm in the appendix C summarizes the overall training procedure of MoSIR.

### 3.6. Theoretical Analysis

We provide theoretical justification for MoSIR by relating the proposed bi-level training objective to a robust surrogate risk, and deriving an OOD generalization bound under distribution shifts. We define the robust surrogate loss as

$$\ell_{\text{rob}}(G, y) = \ell\big(g_\omega(z_G), y\big) + \ell_{\text{sem}}(G) \\ + \lambda_{\text{inv}} \max_{\|\delta\|_2 \leq \epsilon} \mathcal{L}_{\text{inv}}(z_G, \tilde{z}_G). \qquad (18)$$

where $\ell(\cdot, \cdot) \in [0, 1]$ is the task loss after normalization and $\ell_{\text{sem}}(G)$ denotes the semantic consistency regularizer. For simplicity, we write $\ell_{\text{sem}}(G)$ as a per-sample term, while it can be instantiated as a bounded mini-batch regularization.

Let $\mathcal{F}$ denote the hypothesis class induced by MoSIR and define the induced loss class $\mathcal{H} = \{(G, y) \mapsto \ell_{\text{rob}}(G, y) : h \in \mathcal{F}\}$. For any distribution $R$ over $(G, y)$, we define the robust risk $R_R(h) = \mathbb{E}_{(G,y)\sim R}[\ell_{\text{rob}}(G, y)]$.

**Theorem 3.1** (OOD generalization bound (Bartlett & Mendelson, 2002; Ben-David et al., 2010)). *Let $P$ be the training distribution and $Q$ an arbitrary target distribution. For any $\delta \in (0, 1)$, with probability at least $1 - \delta$ over $S \sim P^n$, for all $h \in \mathcal{F}$,*

$$R_Q(h) \leq \widehat{R}_S(h) + 2\mathfrak{R}_S(\mathcal{H}) + 3\sqrt{\frac{\log(2/\delta)}{2n}} \\ + C_{\text{shift}}(P, Q; \mathcal{H}). \qquad (19)$$

*where $\widehat{R}_S(h) = \frac{1}{n}\sum_{i=1}^n \ell_{\text{rob}}(G_i, y_i)$ is the empirical robust risk induced by Eq. (18), $\mathfrak{R}_S(\mathcal{H})$ is the empirical Rademacher complexity of $\mathcal{H}$, and $C_{\text{shift}}(P, Q; \mathcal{H}) = \sup_{g \in \mathcal{H}}\big|\mathbb{E}_Q[g] - \mathbb{E}_P[g]\big|$ measures the discrepancy between $P$ and $Q$ under the robust loss class.*

In particular, Eq. (19) reveals that MoSIR improves OOD generalization by reducing $\mathfrak{R}_S(\mathcal{H})$ via the prototype bottleneck. In addition, enforcing worst-case semantic invariance

under residual perturbations helps mitigate the effect of distribution shifts.

Theorem 3.1 shows that OOD generalization depends on both the complexity term $\mathfrak{R}_S(\mathcal{H})$ and the shift term $C_{\text{shift}}$. MoSIR reduces hypothesis complexity by restricting the predictor to operate on the prototype-projected representation $z_G$, which lies in the convex hull of $\{\mu_k\}_{k=1}^K$.

**Proposition 3.2** (Prototype bottleneck complexity (Massart, 2000)). *Assume $\|\mu_k\|_2 \leq M$ for all $k$ and the prediction head $g_\omega$ is L-Lipschitz. Then the induced loss class satisfies*

$$\mathfrak{R}_S(\mathcal{H}) \leq \mathcal{O}\left(\frac{LM}{\sqrt{n}}\sqrt{\log K}\right). \qquad (20)$$

Proposition 3.2 implies that the prototype bottleneck yields a tighter complexity term in Theorem 3.1, thereby improving OOD generalization. All proofs are provided in Appendix A.

## 4. Experiments

In this section, we conduct extensive experiments to answer the following questions: **Q1:** Does the proposed method improve molecular OOD generalization compared with existing approaches? **Q2:** Can the proposed method learn invariant chemical semantics across distribution shifts? **Q3:** Can the proposed method learn geometry-aware representations that benefit ground-state conformation prediction? **Q4:** How does each component of the proposed method contribute to the overall performance?

### 4.1. Experimental Setup

**Datasets.** Following prior work on OOD molecular representation learning (Zhuang et al., 2023), we perform systematic evaluations on two widely used real-world datasets, namely the GOOD (Gui et al., 2022) and DrugOOD (Ji et al., 2023). **GOOD** contains three molecular datasets: (1)GOOD-HIV is a binary classification task predicting whether a molecule inhibits HIV replication. (2)GOOD-ZINC is a regression task predicting molecular solubility. (3)GOOD-PCBA consists of 128 bioassays and forms 128 binary classification tasks. Each dataset is split into multiple environments using two splitting strategies and two shift types, resulting in twelve OOD settings in total. **DrugOOD** is an OOD benchmark for AI-aided drug discovery. It focuses on drug–target binding affinity prediction under different environment splits. We follow the standard setup and evaluate on six datasets: three splits applied to two measurements, all formulated as binary classification with ROC-AUC as the metric. We evaluate ground-state conformation prediction on two benchmarks. **Molecule3D** (Xu et al., 2021) is a large-scale dataset containing millions of molecules, where the goal is to predict ground-state 3D

*Table 1.* Evaluation performance on the GOOD benchmark. We report ROC-AUC (↑) for GOOD-HIV as it is a binary classification task. For GOOD-ZINC, we report Mean Absolute Error (MAE, ↓) since it is a regression task. For GOOD-PCBA, we report Average Precision (AP, ↑) averaged over all tasks due to severe class imbalance. "–" denotes abnormal results caused by under-fitting as declared in the leaderboard, and "/" indicates that the method is not applicable to the corresponding setting. Boldface indicates the best performance.

| Method | GOOD-HIV ↑ | | | | GOOD-ZINC ↓ | | | | GOOD-PCBA ↑ | | | |
| | scaffold | | size | | scaffold | | size | | scaffold | | size | |
| | covariate | concept | covariate | concept | covariate | concept | covariate | concept | covariate | concept | covariate | concept |
| ERM | 69.55 | 72.48 | 59.19 | 61.91 | 0.1802 | 0.1301 | 0.2319 | 0.1325 | 17.11 | 21.93 | 17.75 | 15.60 |
| IRM | 70.17 | 71.78 | 59.94 | -(-) | 0.2164 | 0.1339 | 0.6984 | 0.1336 | 16.89 | 22.37 | 17.68 | 15.82 |
| VREx | 69.34 | 72.21 | 58.49 | 61.21 | 0.1815 | 0.1287 | 0.2270 | 0.1311 | 17.10 | 21.65 | 17.80 | 15.85 |
| GroupDRO | 68.15 | 71.48 | 57.75 | 59.77 | 0.1870 | 0.1323 | 0.2377 | 0.1333 | 16.55 | 21.91 | 16.74 | 15.21 |
| Coral | 70.69 | 72.96 | 59.39 | 60.29 | 0.1769 | 0.1303 | 0.2292 | 0.1261 | 17.00 | 22.00 | 17.83 | 16.88 |
| DANN | 69.43 | 71.70 | 62.38 | 65.15 | 0.1746 | 0.1269 | 0.2326 | 0.1348 | 17.20 | 22.03 | 17.71 | 15.78 |
| Mixup | 70.65 | 71.89 | 59.11 | 62.80 | 0.2066 | 0.1391 | 0.2531 | 0.1547 | 16.52 | 20.52 | 17.42 | 13.71 |
| DIR | 68.44 | 71.40 | 57.67 | 74.39 | 0.3682 | 0.2543 | 0.4578 | 0.3146 | 16.33 | 23.82 | 16.04 | 16.80 |
| GSAT | 70.07 | 72.51 | 60.73 | 56.96 | 0.1418 | 0.1066 | 0.2101 | 0.1038 | 16.45 | 20.18 | 17.57 | 13.52 |
| GREA | 71.98 | 70.76 | 60.11 | 60.96 | 0.1691 | 0.1157 | 0.2100 | 0.1273 | 16.28 | 20.23 | 17.12 | 13.82 |
| CAL | 69.12 | 72.49 | 59.34 | 56.16 | / | / | / | / | 15.87 | 18.62 | 16.92 | 13.01 |
| DisC | 58.85 | 64.82 | 49.33 | 74.11 | / | / | / | / | / | / | / | / |
| MoleOOD | 69.39 | 69.08 | 58.63 | 55.90 | 0.2752 | 0.1996 | 0.3468 | 0.2275 | 12.90 | 12.92 | 12.64 | 10.30 |
| CIGA | 69.40 | 71.65 | 61.81 | 73.62 | / | / | / | / | / | / | / | / |
| iMoLD | 72.93 | 74.32 | 62.86 | 77.43 | 0.1410 | 0.1014 | 0.1863 | 0.1029 | 17.32 | 22.58 | 18.02 | 18.21 |
| CFD | 76.42 | 77.83 | 64.14 | 79.28 | 0.1187 | 0.0765 | 0.1421 | 0.0852 | 19.78 | 25.64 | 19.18 | 20.03 |
| **MoSIR (Ours)** | **77.13** | **78.96** | **66.65** | **81.15** | **0.0913** | **0.0553** | **0.1317** | **0.0637** | **20.75** | **26.83** | **20.32** | **21.05** |

geometries from 2D molecular graphs under both random and scaffold splits. **QM9** (Ramakrishnan et al., 2014; Wu et al., 2018) is a widely-used quantum chemistry dataset with about 130K small organic molecules, providing DFT-computed stable conformations and quantum properties. More dataset details are provided in Table 7 in the Appendix B.1.

**Baselines and Implementation details.** For OOD molecular representation learning, we compare our method with a broad range of existing approaches. We first consider commonly used methods for non-Euclidean data, including ERM (Vapnik, 2013), IRM (Arjovsky et al., 2019), VREx (Krueger et al., 2021), GroupDRO (Sagawa et al., 2020), Coral (Sun & Saenko, 2016), DANN (Ganin et al., 2016), Mixup (Zhang et al., 2018), as well as the molecule-specific OOD method iMoLD (Zhuang et al., 2023), MILI (Wang et al., 2024) and CFD (Wu & Deng, 2025). These approaches aim to improve robustness by enforcing invariance, reweighting samples, or aligning feature distributions across environments. In addition, we compare our method with graph-specific OOD algorithms designed for molecular data, including CAL (Sui et al., 2022), DisC (Fan et al., 2022), MoleOOD (Yang et al., 2022), and CIGA (Chen et al., 2022). These methods typically mitigate distribution shifts by identifying task-relevant subgraphs or suppressing spurious structural correlations. Finally, we include interpretable graph learning methods such as DIR (Wu et al., 2022b), GSAT (Miao et al., 2022), and GREA (Liu

et al., 2022), which focus on discovering meaningful substructures to enhance robustness and interpretability. We implement the proposed method based on the prior work (Zhuang et al., 2023). For ground-state conformation prediction, we compare with RDKit-based methods DG and ETKDG, DeeperGCN-DAGNN (Xu et al., 2021) for coordinate prediction, strong 2D GNN baselines including GINE (Hu et al., 2019), GATv2 (Brody et al., 2021), and GPS (Rampášek et al., 2022), as well as the transformer-based GTMGC (Xu et al., 2024) and GTMGC + CFD (Wu & Deng, 2025). More implementation details are provided in Table 9 in the Appendix B.4.

### 4.2. Main Results (Q1)

**Results on GOOD.** Table 1 summarizes the performance on the GOOD benchmark. Our method achieves the best results on all three datasets and remains consistently strong across both scaffold/size shifts and covariate/concept settings. Compared with general-purpose OOD baselines such as ERM, IRM, VREx, and GroupDRO, our approach delivers clear improvements, indicating that simply enforcing invariance or reweighting samples is insufficient under systematic molecular distribution shifts. Moreover, our method also surpasses molecule-specific OOD methods such as MoleOOD, CIGA, and iMoLD, especially under concept shift where the structure–property relationship changes across environments. Notably, on GOOD-ZINC, our method attains the lowest MAE, demonstrating its effec-

*Table 2.* ROC-AUC (↑) on DrugOOD. We report IC50 and EC50 under assay-based, scaffold-based, and size-based shifts. Boldface indicates the best performance.

| Method | IC50 ↑ | | | EC50 ↑ | | |
|---|---|---|---|---|---|---|
| | Assay | Scaffold | Size | Assay | Scaffold | Size |
| ERM | 71.63 | 68.79 | 67.50 | 67.39 | 64.98 | 65.10 |
| IRM | 71.15 | 67.22 | 61.58 | 67.77 | 63.86 | 59.19 |
| Coral | 71.28 | 68.36 | 64.53 | 72.08 | 64.83 | 58.47 |
| MixUp | 71.49 | 68.59 | 67.79 | 67.81 | 65.77 | 65.77 |
| DIR | 69.84 | 66.33 | 62.92 | 65.81 | 63.76 | 61.56 |
| GSAT | 70.59 | 66.45 | 66.70 | 73.82 | 64.25 | 62.65 |
| GREA | 70.23 | 67.02 | 66.59 | 74.17 | 64.50 | 62.81 |
| CAL | 70.09 | 65.90 | 66.42 | 74.54 | 65.19 | 61.21 |
| DisC | 61.40 | 62.70 | 61.43 | 63.71 | 60.57 | 57.38 |
| MoleOOD | 71.62 | 68.58 | 65.62 | 72.69 | 65.74 | 65.51 |
| CIGA | 71.86 | 69.14 | 66.92 | 69.15 | 67.32 | 65.65 |
| iMoLD | 72.11 | 68.84 | 67.92 | 77.48 | 67.79 | 67.09 |
| MILI | 72.67 | 69.58 | 68.40 | 77.11 | 68.07 | 65.97 |
| CFD | 73.86 | 70.02 | 69.73 | 78.32 | 69.13 | 67.62 |
| **MoSIR (Ours)** | **74.65** | **71.50** | **72.03** | **79.52** | **70.25** | **68.56** |

tiveness for both classification and regression tasks.

**Results on DrugOOD.**   Results on DrugOOD are reported in Table 2. Our method consistently achieves the highest ROC-AUC on both IC50 and EC50 prediction under assay-based, scaffold-based, and size-based shifts, showing robustness across diverse distribution shift mechanisms. In particular, the improvements under scaffold and size shifts suggest that our method effectively mitigates environment-dependent spurious correlations induced by structural variations. In addition, our approach remains superior to strong molecule OOD baselines, including MoleOOD, CIGA, iMoLD, MILI, and CFD, highlighting its advantage in realistic drug discovery scenarios. Overall, these results confirm that semantic prototype decomposition combined with worst-case distribution modeling leads to more invariant and transferable molecular representations.

### 4.3. Invariant Chemical Semantics (Q2)

To verify whether the learned prototype space captures meaningful chemical semantics, we analyze the prototypes from four perspectives, including functional group selectivity, substructure molecule retrieval, quantitative substructure consistency, and physicochemical property profiling.

**Functional group selectivity.**   Using RDKit SMARTS patterns, we detect $F = 8$ functional groups. For each molecule $G$, let $p_G(k)$ denote its prototype activation. We define the selectivity of prototype $k$ to functional group $r$ as

$$A_{k,r} = \mathbb{E}[p_G(k) \mid c_G(r) = 1] - \mathbb{E}[p_G(k) \mid c_G(r) = 0], \tag{21}$$

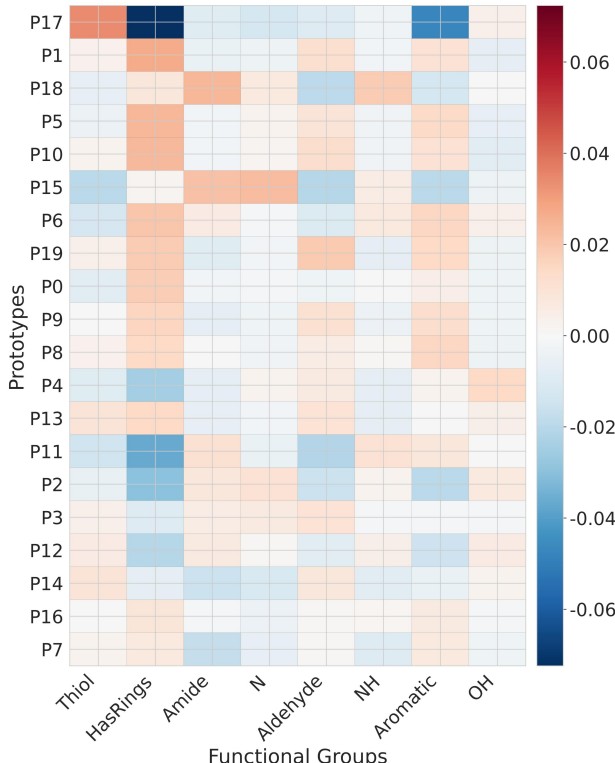

*Figure 2.* Prototype–functional group selectivity heatmap. Each entry $A_{k,r}$ measures the difference between the expected activation of prototype $k$ when functional group $r$ is present and absent. Red (blue) indicates positive (negative) selectivity, meaning that the presence of $r$ increases (suppresses) the activation of prototype $k$.

where $c_G(f)$ indicates whether $G$ contains $r$. Figure 2 visualizes $A_{k,f}$ as a heatmap. The sparse and structured patterns show that different prototypes specialize in distinct chemical motifs, such as rings, heterocycles, and hydrogen-bond related groups.

**Substructure molecule retrieval.**   We qualitatively validate the learned representations by retrieving the top-activated molecules for each prototype. Figure 3 displays the results for selected prototypes along with the comprehensive visualization in the Figure 4 Appendix in the Appendix B.3. The results reveal that molecules activated by the same prototype share highly consistent substructures, demonstrating that the prototypes have effectively captured invariant chemical semantics. This consistency is crucial for OOD generalization as it suggests the model relies on stable functional determinants rather than spurious environmental correlations, thereby ensuring reliable predictions across diverse chemical environments.

**Quantitative substructure consistency.**   We further conduct a quantitative substructure consistency analysis on GOOD-ZINC under size and concept shifts. For each

*Table 3.* Quantitative structural consistency of learned prototypes on GOOD-ZINC under size and concept shifts. For each prototype, we select its top-activated molecules and compute the mean intra-prototype pairwise Tanimoto similarity based on Morgan fingerprints. Random controls are constructed by sampling molecule sets of the same size over 50 runs. $\Delta$ denotes the difference between intra-prototype and random-control similarities.

| High-Activation Rule | Mean Intra-Prototype Tanimoto | Mean Random-Control Tanimoto | $\Delta$ (Intra $-$ Random) | 95% CI of $\Delta$ | % Prototypes with $\Delta > 0$ | % Prototypes with $p < 0.05$ |
|---|---|---|---|---|---|---|
| Top 5% activation | 0.1296 | 0.1170 | +0.0126 | [0.0056, 0.0198] | 75.0% | 65.0% |
| Top 10% activation | 0.1272 | 0.1170 | +0.0102 | [0.0031, 0.0173] | 65.0% | 65.0% |
| Top 20% activation | 0.1242 | 0.1170 | +0.0072 | [0.0012, 0.0135] | 65.0% | 45.0% |

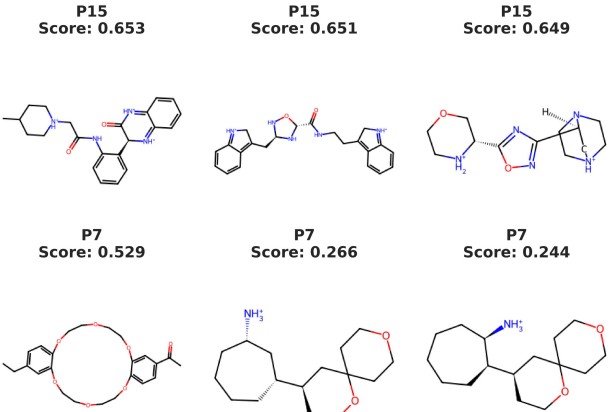

*Figure 3.* Top-activated molecules for representative semantic prototypes. For each prototype, we visualize the molecules with the highest activation scores. Molecules associated with the same prototype exhibit highly consistent substructure and functional group patterns.

molecule, we compute Morgan fingerprints and measure structural similarity using the Tanimoto coefficient. For each prototype, we select the top 5%, 10%, and 20% activated molecules, compute their average pairwise Tanimoto similarity, and compare it with random molecule sets of the same size. Random controls are averaged over 50 runs, and bootstrap 95% confidence intervals are reported for the similarity difference. As shown in Table 3, high-activation molecules within the same prototype consistently show higher structural similarity than random controls across all thresholds. The gain is larger under stricter activation rules, with the top 5% molecules achieving the strongest improvement. These results quantitatively support that the learned prototype space captures structurally consistent chemical semantics rather than random molecular clusters.

**Physicochemical property profiling.** We compute six RDKit descriptors including MW, logP, NumRings, HBD, HBA, and TPSA and assign each molecule to its top-activated prototype. As illustrated in Figure 5 in the Appendix B.3, different prototypes exhibit clearly shifted property distributions, indicating that the prototype space partitions molecules into distinct physicochemical subspaces.

### 4.4. Ground-state Conformation Prediction (Q3)

We further evaluate MoSIR on ground-state conformation prediction as an auxiliary task to examine whether the learned representations capture chemically meaningful substructure information. This task requires predicting stable 3D molecular geometries from 2D molecular graphs, and therefore provides an additional evaluation of whether the learned representations encode structural factors related to molecular geometry. Following prior work (Xu et al., 2024), we conduct experiments on Molecule3D and QM9, and report D-MAE, D-RMSE, and C-RMSD. We follow the standard experimental setup and keep the backbone architecture and training hyperparameters unchanged, where MoSIR is applied as a plug-in module. As shown in Table 4, MoSIR consistently improves upon the transformer-based GTMGC backbone across all reported metrics on both validation and test sets. The reductions in D-MAE, D-RMSE, and C-RMSD indicate that MoSIR improves both distance-level and conformation-level geometry prediction. These results further confirm that the prototype-based semantic bottleneck helps preserve chemically meaningful substructure information and benefits 3D molecular geometry modeling.

### 4.5. Ablation Studies (Q4)

We conduct ablation studies on the GOOD-ZINC dataset to examine the effect of different loss in the proposed objective. As reported in Table 5, removing either the semantic consistency loss $\mathcal{L}_{\text{sem}}$ or the semantic invariant loss $\mathcal{L}_{\text{inv}}$ consistently degrades OOD performance. These results demonstrate that enforcing semantic stability and robustness against worst-case perturbations is crucial for learning invariant and transferable molecular representations.

### 4.6. Hyperparameter Analysis (Q4)

We analyze the sensitivity of our method to the number of semantic prototypes $K$ on the GOOD-ZINC dataset. Specifically, we vary $K \in \{5, 10, 20, 30, 40\}$ while keeping all other hyperparameters fixed. Table 6 shows that when $K$ is too small, the prototype dictionary has limited capacity to capture diverse chemical semantics, leading to suboptimal performance. As $K$ increases, performance consistently

*Table 4.* Performance of ground-state conformation prediction on Molecule3D and QM9. We report D-MAE, D-RMSE, and C-RMSD for both validation and test sets. Lower is better. Bold indicates the best performance.

| Method | Validation | | | Test | | |
|---|---|---|---|---|---|---|
| | D-MAE↓ | D-RMSE↓ | C-RMSD↓ | D-MAE↓ | D-RMSE↓ | C-RMSD↓ |
| (a) Molecule3D (Random Split) | | | | | | |
| RDKit DG | 0.581 | 0.930 | 1.054 | 0.582 | 0.932 | 1.055 |
| RDKit ETKDG | 0.575 | 0.941 | 0.998 | 0.576 | 0.942 | 0.999 |
| DeeperGCN-DAGNN | 0.509 | 0.849 | – | 0.571 | 0.961 | – |
| GINE | 0.590 | 1.014 | 1.116 | 0.592 | 1.018 | 1.116 |
| GATv2 | 0.563 | 0.983 | 1.082 | 0.564 | 0.986 | 1.083 |
| GPS | 0.528 | 0.909 | 1.036 | 0.529 | 0.911 | 1.038 |
| GTMGC | 0.432 | 0.719 | 0.712 | 0.433 | 0.721 | 0.713 |
| GTMGC + CFD | 0.397 | 0.682 | 0.684 | 0.407 | 0.695 | 0.688 |
| **GTMGC + MoSIR (Ours)** | **0.357** | **0.655** | **0.659** | **0.385** | **0.663** | **0.652** |
| (b) QM9 | | | | | | |
| RDKit DG | 0.358 | 0.616 | 0.722 | 0.358 | 0.615 | 0.722 |
| RDKit ETKDG | 0.355 | 0.621 | 0.691 | 0.355 | 0.621 | 0.689 |
| GINE | 0.357 | 0.673 | 0.685 | 0.357 | 0.669 | 0.693 |
| GATv2 | 0.339 | 0.663 | 0.661 | 0.339 | 0.659 | 0.666 |
| GPS | 0.326 | 0.644 | 0.662 | 0.326 | 0.640 | 0.666 |
| GTMGC | 0.262 | 0.468 | 0.362 | 0.264 | 0.470 | 0.367 |
| GTMGC + CFD | 0.223 | 0.434 | 0.305 | 0.218 | 0.442 | 0.309 |
| **GTMGC + MoSIR (Ours)** | **0.205** | **0.410** | **0.256** | **0.187** | **0.415** | **0.258** |

*Table 5.* Ablation studies on GOOD-ZINC (MAE ↓). $\mathcal{L}_{\text{sem}}$ denotes the semantic consistency loss, and $\mathcal{L}_{\text{inv}}$ denotes the semantic invariant loss.

| Model | Scaffold | | Size | |
|---|---|---|---|---|
| | covariate | concept | covariate | concept |
| MoSIR (Ours) | **0.0913** | **0.0553** | **0.1317** | **0.0637** |
| w/o $\mathcal{L}_{\text{sem}}$ | 0.0986 | 0.0617 | 0.1413 | 0.0712 |
| w/o $\mathcal{L}_{\text{inv}}$ | 0.0959 | 0.0589 | 0.1378 | 0.0674 |

*Table 6.* Sensitivity analysis on the number of prototypes $K$ on GOOD-ZINC (MAE ↓). We report results under scaffold/size shifts with both covariate and concept settings. Boldface indicates the best performance.

| $K$ | Scaffold | | Size | |
|---|---|---|---|---|
| | covariate | concept | covariate | concept |
| 5 | 0.1089 | 0.0678 | 0.1524 | 0.0819 |
| 10 | 0.0997 | 0.0612 | 0.1410 | 0.0725 |
| 20 | **0.0913** | **0.0553** | **0.1317** | **0.0637** |
| 30 | 0.0946 | 0.0578 | 0.1359 | 0.0668 |
| 40 | 0.0951 | 0.0581 | 0.1364 | 0.0672 |

improves and reaches the best results at a moderate value (i.e., $K = 20$). However, using an excessively large $K$ introduces redundant prototypes and slightly destabilizes optimization, resulting in marginal gains or mild performance degradation.

## 5. Conclusion

We studied out-of-distribution generalization in molecular representation learning, where models often fail under scaffold, size, and environment shifts due to spurious environment-dependent correlations. To address this issue, we proposed MoSIR, a semantic invariant learning framework that decomposes molecular representations into prototype-based semantic components and environment-dependent residual factors. By projecting entangled molec-

ular embeddings into a learnable semantic prototype space, MoSIR encourages the model to rely on stable chemical semantics rather than shift-sensitive structural variations. MoSIR further introduces adversarial distribution modeling in the residual space and enforces semantic stability through a bi-level min–max optimization process. Extensive experiments on GOOD and DrugOOD demonstrate that MoSIR consistently outperforms strong baselines across diverse OOD settings. Additional analyses show that the learned prototypes encode meaningful and structurally consistent chemical semantics. These results highlight the importance of learning semantic invariant representations and provide a robust foundation for reliable molecular property prediction under distribution shifts.

## Acknowledgements

This work is supported by the National Natural Science Foundation of China (U25A20529, 62376142, 62272285). We extend our heartfelt gratitude to the anonymous reviewers for their invaluable and constructive comments.

## Impact Statement

This paper presents work whose goal is to advance the field of Machine Learning. There are many potential societal consequences of our work, none which we feel must be specifically highlighted here.

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

# A. Proofs for Theoretical Analysis

In this section, we provide the detailed proofs for the theoretical results presented in Section 3.6.

## A.1. Preliminaries and Definitions

We first review the definition of Rademacher complexity. Let $\mathcal{X}$ be the input space and $\mathcal{D}$ be a distribution over $\mathcal{X}$. Let $\mathcal{F}$ be a class of real-valued functions defined on $\mathcal{X}$. Given a sample $S = \{x_1, \ldots, x_n\}$ drawn i.i.d. from $\mathcal{D}$, the *empirical Rademacher complexity* of $\mathcal{F}$ with respect to $S$ is defined as:

$$\mathfrak{R}_S(\mathcal{F}) = \mathbb{E}_{\boldsymbol{\sigma}} \left[ \sup_{f \in \mathcal{F}} \frac{1}{n} \sum_{i=1}^n \sigma_i f(x_i) \right], \tag{1}$$

where $\boldsymbol{\sigma} = (\sigma_1, \ldots, \sigma_n)$ are independent Rademacher variables with $P(\sigma_i = 1) = P(\sigma_i = -1) = 0.5$.

## A.2. Proof of Theorem 3.1 (OOD Generalization Bound)

**Theorem 3.1.** *Let $P$ be the training distribution and $Q$ be an arbitrary target distribution. For any $\delta \in (0, 1)$, with probability at least $1 - \delta$ over sample $S \sim P^n$, for all hypotheses $h \in \mathcal{F}$:*

$$R_Q(h) \le \widehat{R}_S(h) + 2\mathfrak{R}_S(\mathcal{H}) + 3\sqrt{\frac{\log(2/\delta)}{2n}} + C_{\text{shift}}(P, Q; \mathcal{H}).$$

*Proof.* The proof relies on decomposing the risk on the target distribution $Q$ into terms involving the empirical risk on the source distribution $P$ and the distribution discrepancy.

First, we decompose the true risk on the target distribution $R_Q(h)$ as:

$$R_Q(h) = R_P(h) + (R_Q(h) - R_P(h)). \tag{2}$$

By the definition of the shift measure $C_{\text{shift}}(P, Q; \mathcal{H})$ given in the theorem:

$$R_Q(h) - R_P(h) = \mathbb{E}_Q[\ell_{\text{rob}}] - \mathbb{E}_P[\ell_{\text{rob}}] \le \sup_{g \in \mathcal{H}} |\mathbb{E}_Q[g] - \mathbb{E}_P[g]| = C_{\text{shift}}(P, Q; \mathcal{H}). \tag{3}$$

Thus,

$$R_Q(h) \le R_P(h) + C_{\text{shift}}(P, Q; \mathcal{H}). \tag{4}$$

Now we bound $R_P(h)$ using the empirical risk $\widehat{R}_S(h)$. Since the loss function $\ell_{\text{rob}}$ is bounded in $[0, 1]$ (as $\ell(\cdot, \cdot) \in [0, 1]$ and the regularization terms are assumed bounded/normalized), we can apply standard uniform convergence bounds based on Rademacher complexity.

According to the standard generalization bound (e.g., Theorem 3.3 in (Mohri et al., 2018)), for any $\delta \in (0, 1)$, with probability at least $1 - \delta$ over $S \sim P^n$, we have:

$$\sup_{h \in \mathcal{F}}(R_P(h) - \widehat{R}_S(h)) \le 2\mathfrak{R}_S(\mathcal{H}) + 3\sqrt{\frac{\log(2/\delta)}{2n}}. \tag{5}$$

Substituting the bound from Step 2 into the decomposition from Step 1, we obtain:

$$R_Q(h) \le \widehat{R}_S(h) + 2\mathfrak{R}_S(\mathcal{H}) + 3\sqrt{\frac{\log(2/\delta)}{2n}} + C_{\text{shift}}(P, Q; \mathcal{H}). \tag{6}$$

This concludes the proof. $\square$

### A.3. Proof of Proposition 3.2 (Prototype Bottleneck Complexity)

**Proposition 3.2.** *Assume $\|\mu_k\|_2 \leq M$ for all $k$ and the prediction head $g_\omega$ is $L$-Lipschitz. Then the empirical Rademacher complexity satisfies:*

$$\mathfrak{R}_S(\mathcal{H}) \leq \mathcal{O}\left(\frac{LM}{\sqrt{n}}\sqrt{\log K}\right).$$

*Proof.* Let the hypothesis class be defined by the composition of the semantic projection and the prediction head. Specifically, the core term in the loss is $\ell(g_\omega(z_G), y)$. We focus on bounding the complexity of the function class induced by the semantic invariant representation $z_G$.

Recall that in MoSIR, the semantic representation $z_G$ is computed as a probability-weighted sum of prototypes:

$$z_G = \sum_{k=1}^{K} p(k|u_G)\mu_k, \quad \text{where} \sum_{k=1}^{K} p(k|u_G) = 1, \quad p(k|u_G) \geq 0. \tag{7}$$

Geometrically, this implies that for any graph $G$, the representation $z_G$ lies in the convex hull of the prototype set $\mathcal{M} = \{\mu_1, \ldots, \mu_K\}$. Let $\mathcal{Z} = \text{conv}(\mathcal{M})$ denote this convex hull.

We assume the loss function $\ell$ is 1-Lipschitz (or bounded by a constant absorbed into $L$) with respect to the prediction, and the prediction head $g_\omega$ is $L$-Lipschitz with respect to $z_G$. By Talagrand's Contraction Lemma (Lemma 5.7 in (Mohri et al., 2018)), we can peel off the Lipschitz functions:

$$\mathfrak{R}_S(\mathcal{H}) \leq L \cdot \mathfrak{R}_S(\mathcal{Z}), \tag{8}$$

where $\mathfrak{R}_S(\mathcal{Z})$ is the Rademacher complexity of the set of possible semantic representations $\{z_{G_1}, \ldots, z_{G_n}\} \subseteq \mathcal{Z}$.

The crucial step relies on the property of Rademacher complexity for convex hulls. For any finite set of points $A$, the Rademacher complexity of its convex hull is equal to the Rademacher complexity of the set itself:

$$\mathfrak{R}_S(\text{conv}(\mathcal{M})) = \mathfrak{R}_S(\mathcal{M}). \tag{9}$$

Therefore, we only need to bound the complexity of the discrete set of $K$ prototypes $\mathcal{M} = \{\mu_1, \ldots, \mu_K\}$. Using Massart's Finite Class Lemma (Massart, 2000), for a finite hypothesis set $\mathcal{M}$ with cardinality $K$ and maximum $L_2$-norm bounded by $M$ (i.e., $\max_k \|\mu_k\|_2 \leq M$):

$$\mathfrak{R}_S(\mathcal{M}) \leq M\sqrt{\frac{2\log K}{n}}. \tag{10}$$

Combining these results:

$$\mathfrak{R}_S(\mathcal{H}) \leq L \cdot \mathfrak{R}_S(\mathcal{Z}) = L \cdot \mathfrak{R}_S(\mathcal{M}) \leq L \cdot M\sqrt{\frac{2\log K}{n}}. \tag{11}$$

Using the Big-O notation to hide constants, we arrive at:

$$\mathfrak{R}_S(\mathcal{H}) \leq \mathcal{O}\left(\frac{LM}{\sqrt{n}}\sqrt{\log K}\right). \tag{12}$$

This proves that the complexity is logarithmically dependent on the number of prototypes $K$, justifying the effectiveness of the prototype bottleneck. $\square$

## B. Additional Experimental Details

### B.1. Dataset Details

**Benchmarks and Task Overview.** We evaluate out-of-distribution molecular representation learning on two public benchmarks, namely **GOOD** and **DrugOOD**. In all experiments, each molecule is represented as a molecular graph. The downstream tasks include binary classification evaluated by ROC-AUC, regression evaluated by mean absolute error, and multi-task binary classification evaluated by average precision. For OOD evaluation, we strictly follow the official data splits of each benchmark, where training, validation, and test sets are separated according to environment partitions induced by different splitting strategies, as summarized in Table 7.

*Table 7.* Statistics of datasets and evaluation settings used in this work. For each benchmark, we report the task type, evaluation metric, environment split strategy, and the number of training, validation, and test samples. The GOOD benchmark includes covariate and concept shifts under scaffold and size splits, covering binary classification, regression, and multi-task classification tasks. The DrugOOD benchmark focuses on covariate shifts under assay, scaffold, and size splits for molecular activity prediction.

| Dataset | Split / Shift | Task | Metric | #Train | #Val | #Test | #Tasks |
|---|---|---|---|---|---|---|---|
| GOOD-HIV | scaffold / covariate | Binary Classification | ROC-AUC | 24682 | 4133 | 4108 | 1 |
| | scaffold / concept | Binary Classification | ROC-AUC | 15209 | 9365 | 10037 | 1 |
| | size / covariate | Binary Classification | ROC-AUC | 26169 | 4112 | 3961 | 1 |
| | size / concept | Binary Classification | ROC-AUC | 14454 | 3096 | 10525 | 1 |
| GOOD-ZINC | scaffold / covariate | Regression | MAE | 149674 | 24945 | 24946 | 1 |
| | scaffold / concept | Regression | MAE | 101867 | 43539 | 60393 | 1 |
| | size / covariate | Regression | MAE | 161893 | 24945 | 17042 | 1 |
| | size / concept | Regression | MAE | 89418 | 19161 | 70306 | 1 |
| GOOD-PCBA | scaffold / covariate | Multi-task Binary Classification | AP | 262764 | 44019 | 43562 | 128 |
| | scaffold / concept | Multi-task Binary Classification | AP | 159158 | 90740 | 119821 | 128 |
| | size / covariate | Multi-task Binary Classification | AP | 269990 | 43792 | 31925 | 128 |
| | size / concept | Multi-task Binary Classification | AP | 150121 | 32168 | 115205 | 128 |
| DrugOOD-IC50 | assay / covariate | Binary Classification | ROC-AUC | 34953 | 19475 | 19463 | 1 |
| | scaffold / covariate | Binary Classification | ROC-AUC | 22025 | 19478 | 19480 | 1 |
| | size / covariate | Binary Classification | ROC-AUC | 37497 | 17987 | 16761 | 1 |
| DrugOOD-EC50 | assay / covariate | Binary Classification | ROC-AUC | 4978 | 2761 | 2725 | 1 |
| | scaffold / covariate | Binary Classification | ROC-AUC | 2743 | 2723 | 2762 | 1 |
| | size / covariate | Binary Classification | ROC-AUC | 5189 | 2495 | 2505 | 1 |

**GOOD benchmark.** The GOOD benchmark (Gui et al., 2022) consists of three datasets, GOOD-HIV, GOOD-ZINC, and GOOD-PCBA. It evaluates two types of distribution shifts, namely covariate shift and concept shift, under two environment splitting strategies, including scaffold-based splits and size-based splits. Combining three datasets with two shift types and two splitting strategies results in a total of twelve evaluation settings.

**DrugOOD benchmark.** The DrugOOD benchmark (Ji et al., 2023) focuses on covariate shift and includes six datasets derived from two measurement types, IC50 and EC50. Each measurement type is further divided into three environment splits, including assay-based, scaffold-based, and size-based splits. We adopt the default data splits and evaluation protocol provided by the benchmark.

**Molecule3D.** Molecule3D (Xu et al., 2021) is a large-scale benchmark for ground-state conformation prediction, which aims to predict the lowest-energy 3D molecular geometries based solely on 2D molecular graph structures. It contains approximately 4 million molecules, each associated with a 2D molecular graph, a ground-state 3D conformation, and additional quantum properties. Following the standard protocol, we evaluate under the *random split*, where molecules are randomly partitioned according to the same distribution, using a 6:2:2 ratio for training, validation, and testing.

**QM9.** QM9 (Ramakrishnan et al., 2014; Wu et al., 2018) is a quantum chemistry dataset that provides geometry, energy, electronic, and thermodynamic properties for nearly 130,000 small organic molecules with up to 9 heavy atoms. Each molecule is associated with its most stable conformation computed by density functional theory (DFT). We follow the data split in Liao & Smidt (2022), using 110K, 10K, and 11K molecules for training, validation, and testing, respectively.

### B.2. What Causes Distribution Shifts?

Out-of-distribution generalization in molecular learning primarily arises from changes in the *environment* from which molecules are sampled. Different environments induce systematic variations in molecular distributions, even when the prediction task remains unchanged. In this work, we operationalize environments using three widely adopted splitting strategies: **scaffold**, **size**, and **assay**. These strategies correspond to distinct scientific sources of distribution shift, including changes in core chemical motifs, molecular scale, and experimental measurement conditions. An overview of the environment definitions and the associated sources of distribution shift is provided in Table 8.

*Table 8.* Environment definitions and corresponding sources of molecular distribution shifts.

| Environment | Varying Factor | Fixed Factor | Source of Distribution Shift |
|---|---|---|---|
| Scaffold | Core chemical motif | Molecular size and assay | Change in fundamental structural backbone |
| Size | Number of atoms | Chemical motif and assay | Change in molecular scale and complexity |
| Assay | Measurement condition | Molecular structure | Change in experimental setup and target |

### B.3. Additional Qualitative Analysis of Semantic Prototypes

This section provides additional qualitative analyses to complement the quantitative OOD evaluation reported in the main paper. Beyond predictive performance, we investigate whether the learned prototype space exhibits interpretable chemical semantics. To this end, we visualize the top-activated molecules for representative prototypes and examine the physicochemical property distributions across prototypes, as shown in Figures 4 and 5. The top-activated molecule visualization assesses whether molecules assigned to the same prototype share consistent substructures or functional motifs. The physicochemical property profiling further evaluates whether different prototypes correspond to distinguishable molecular property regions, including molecular weight, logP, ring count, hydrogen-bond donors and acceptors, and topological polar surface area. These analyses show that the learned prototypes are not merely optimization artifacts, but instead capture invariant and interpretable chemical semantics that are consistent with molecular structure and properties.

### B.4. Additional Implementation Details

Table 9 reports the detailed hyperparameter configurations for all datasets. These hyperparameters include the neighborhood size $|\mathcal{N}|$, the number of semantic prototypes $K$, loss balancing coefficients, and optimization-related parameters.

*Table 9.* Detailed hyperparameter configurations for all datasets under different distribution shifts.

| Dataset | Shift | $|\mathcal{N}|$ | $K$ | batch-size | $\lambda_{\text{intra}}$ | $\lambda_{\text{inter}}$ | $\lambda_{\text{inv}}$ | dropout |
|---|---|---|---|---|---|---|---|---|
| GOOD-HIV | scaffold / covariate | 5 | 30 | 256 | 0.3 | 0.3 | 1 | 0.5 |
| | scaffold / concept | 10 | 30 | 1024 | 0.3 | 0.3 | 1 | 0.5 |
| | size / covariate | 8 | 20 | 256 | 0.3 | 0.3 | 1 | 0.5 |
| | size / concept | 8 | 30 | 256 | 0.3 | 0.3 | 1 | 0.5 |
| GOOD-ZINC | scaffold / covariate | 8 | 20 | 64 | 0.002 | 0.1 | 0.1 | 0 |
| | scaffold / concept | 10 | 20 | 256 | 0.002 | 0.1 | 0.1 | 0 |
| | size / covariate | 8 | 20 | 64 | 0.002 | 0.1 | 0.1 | 0 |
| | size / concept | 10 | 20 | 64 | 0.002 | 0.1 | 0.1 | 0 |
| GOOD-PCBA | scaffold / covariate | 5 | 20 | 32 | 0.3 | 0.3 | 1 | 0.5 |
| | scaffold / concept | 5 | 30 | 32 | 0.3 | 0.3 | 1 | 0.5 |
| | size / covariate | 8 | 20 | 32 | 0.3 | 0.3 | 1 | 0.5 |
| | size / concept | 10 | 30 | 32 | 0.3 | 0.3 | 1 | 0.5 |
| DrugOOD-IC50 | assay / covariate | 8 | 30 | 128 | 0.002 | 0.1 | 0.1 | 0.5 |
| | scaffold / covariate | 5 | 30 | 128 | 0.002 | 0.1 | 0.1 | 0.1 |
| | size / covariate | 8 | 20 | 128 | 0.002 | 0.1 | 0.1 | 0.5 |
| DrugOOD-EC50 | assay / covariate | 8 | 20 | 128 | 0.002 | 0.1 | 0.1 | 0.5 |
| | scaffold / covariate | 8 | 30 | 128 | 0.002 | 0.1 | 0.1 | 0.3 |
| | size / covariate | 10 | 20 | 128 | 0.002 | 0.1 | 0.1 | 0.5 |

## C. Algorithmic Description of MoSIR

For completeness and reproducibility, we provide a compact algorithmic description of MoSIR. The algorithm summarizes the training pipeline, including graph encoding, prototype-based semantic decomposition, residual perturbation, and joint optimization. For each mini-batch, MoSIR learns semantic invariant representations through prototype projection and enforces their stability under residual-space perturbations. This gives a clear implementation-level overview of the proposed framework.

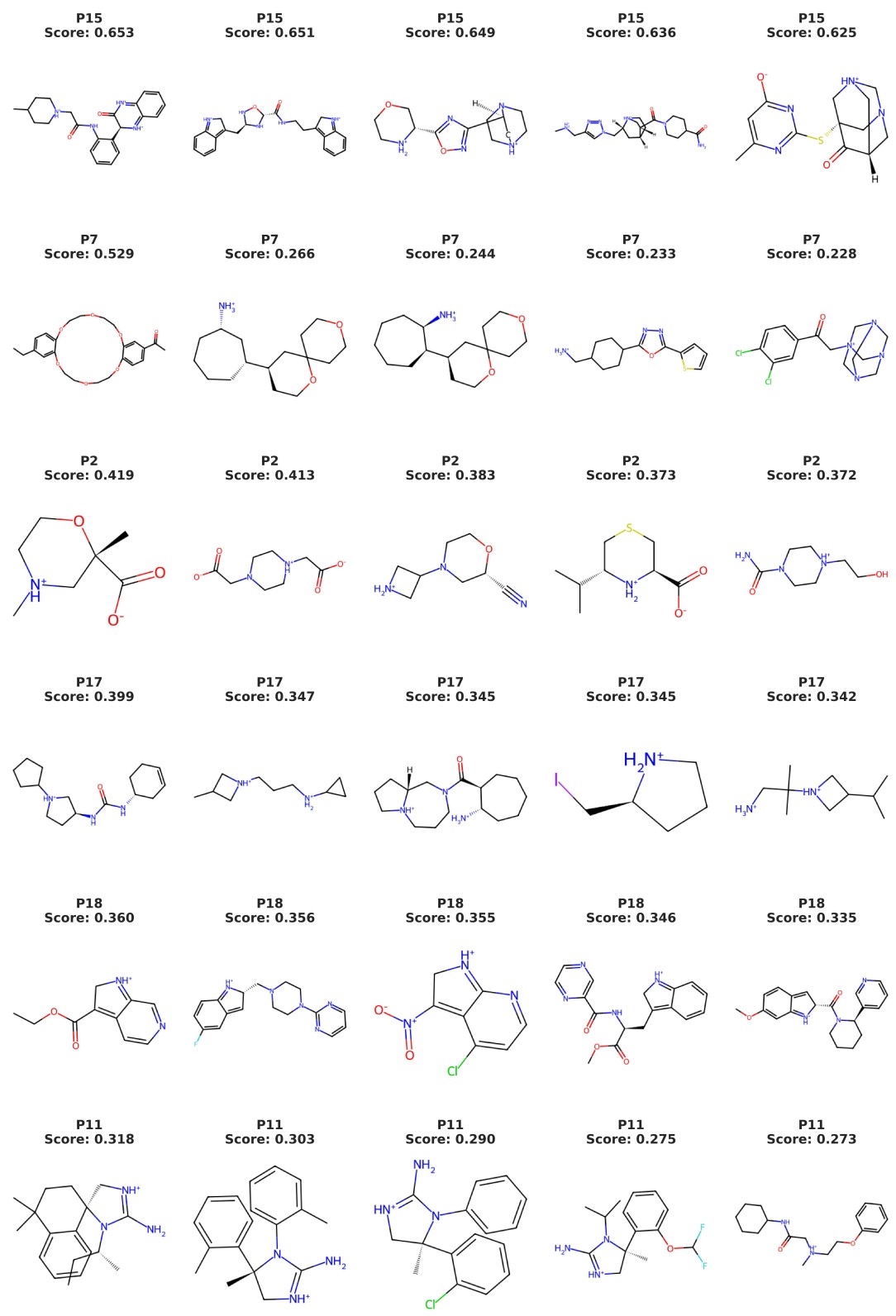

*Figure 4.* Visualization of top-activated molecules for representative semantic prototypes. For selected prototypes including $P_{15}$, $P_7$, $P_2$, $P_{17}$, $P_{18}$, and $P_{11}$, we display the top-5 molecules with the highest activation scores from the dataset. The high structural similarity within each group evidenced by shared scaffolds or functional motifs demonstrates that the learned prototypes successfully capture consistent and interpretable chemical semantics.

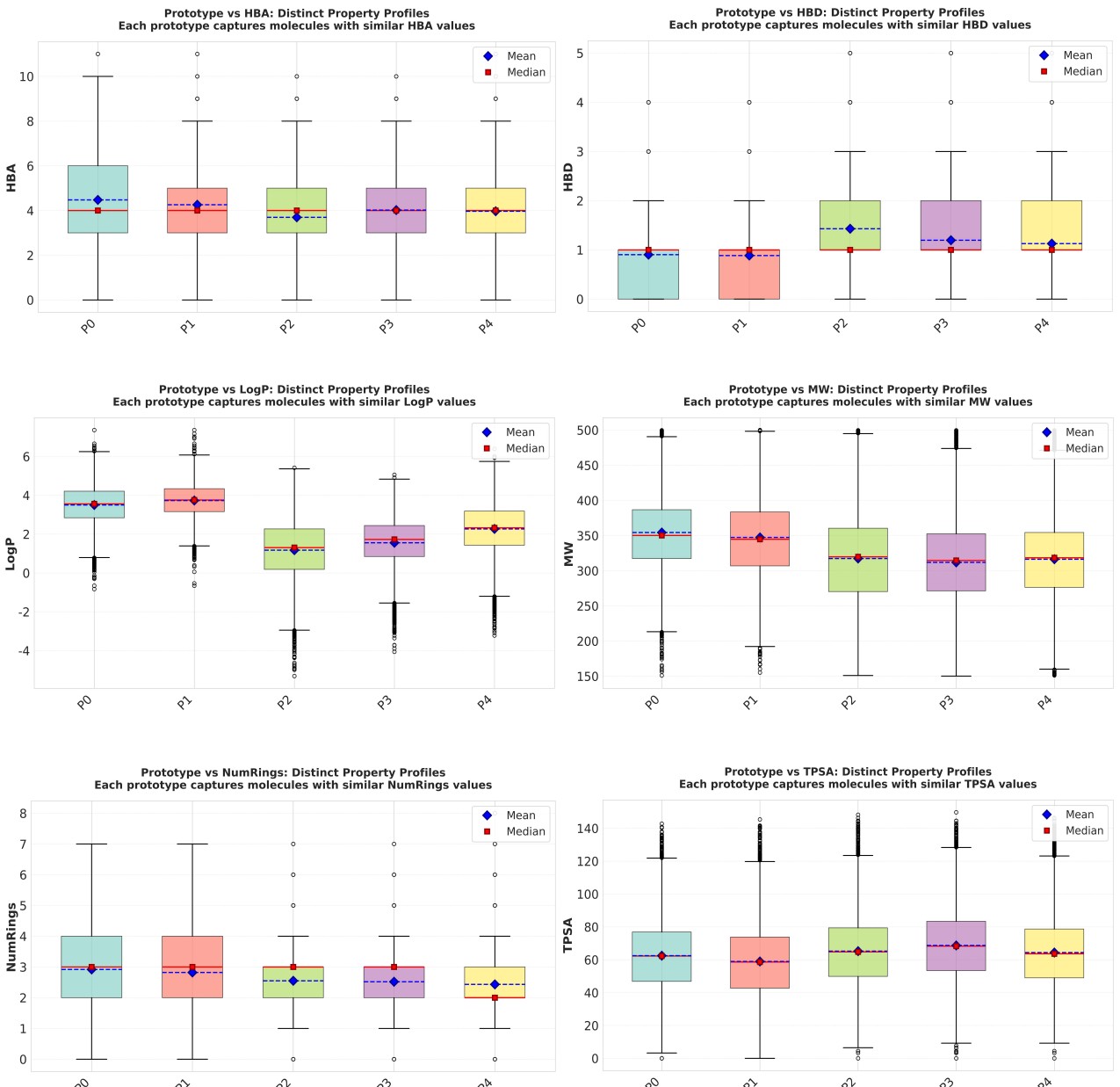

*Figure 5.* Interpretability analysis of the learned semantic prototypes. The box-and-whisker plots visualize the distributions of six key physicochemical properties across the learned prototypes $P_0$ through $P_4$. The distinct property profiles demonstrate that MoSIR successfully disentangles molecular graphs into meaningful semantic subspaces without explicit supervision. For instance, $P_2$ captures hydrophilic semantics characterized by low LogP and high HBD, while $P_1$ exhibits strong lipophilic characteristics indicated by high LogP and low TPSA, validating that the learned prototypes encode interpretable chemical concepts.

---

**Algorithm 1** MoSIR: Training Procedure

---

**Require:** Training data $\mathcal{D}$; encoder $f_\theta$; prototypes $\{\mu_k\}_{k=1}^K$; perturbation radius $\epsilon$; loss weights $\lambda_{\text{intra}}, \lambda_{\text{inter}}, \lambda_{\text{inv}}$
**Ensure:** Trained $f_\theta$, $\{\mu_k\}$, predictor $g_\omega$
 1: Initialize parameters $\theta$, $\{\mu_k\}_{k=1}^K$, and $\omega$
 2: **while** not converged **do**
 3:  Sample mini-batch $\mathcal{B} = \{(G, y)\}$
 4:  **Semantic Prototype Decomposition**
 5:  Compute $u_G = \text{Pool}(f_\theta(G))$
 6:  Compute prototype assignment $p(k|u_G) = \text{softmax}(\mu_k^\top u_G / \tau)$
 7:  Compute semantic and residual representations: $z_G = \sum_k p(k|u_G)\mu_k, \quad e_G = u_G - z_G$
 8:  **Semantic Consistency Constraints**
 9:  $\mathcal{L}_{\text{intra}} = \frac{1}{|\mathcal{B}|} \sum_{G \in \mathcal{B}} \|u_G - z_G\|_2^2 + \lambda_\mu \sum_k \|\mu_k\|_2^2,$
10:  $\mathcal{L}_{\text{inter}} = \frac{1}{|\mathcal{B}|} \sum_{G \in \mathcal{B}} \sum_{G' \in \mathcal{N}(G)} w_{GG'} \|z_G - z_{G'}\|_2^2$
11:  $\mathcal{L}_{\text{sem}} = \lambda_{\text{intra}}\mathcal{L}_{\text{intra}} + \lambda_{\text{inter}}\mathcal{L}_{\text{inter}}$
12:  **Residual adversarial perturbation**
13:  Initialize $\delta \sim \mathcal{U}(-\epsilon, \epsilon)$
14:  **for** $t = 1$ to $T$ **do**
15:    $\tilde{u}_G = \text{sg}(z_G) + \text{sg}(e_G) + \delta$
16:    $\tilde{z}_G = \sum_k p(k|\tilde{u}_G)\mu_k$
17:    $\delta \leftarrow \Pi_{\|\delta\| \leq \epsilon}(\delta + \eta_\delta \nabla_\delta \mathcal{L}_{\text{inv}}(z_G, \tilde{z}_G))$
18:  **end for**
19:  **Invariant learning and update**
20:  Compute $\mathcal{L}_{\text{inv}}(z_G, \tilde{z}_G)$
21:  Update $\theta, \{\mu_k\}, \omega$ by minimizing $\mathcal{L}_{\text{task}} + \mathcal{L}_{\text{sem}} + \lambda_{\text{inv}}\mathcal{L}_{\text{inv}}$
22: **end while**

---

