# OpenReview forum: "Learning Molecular Semantic Invariant Representation with Prototype Constraint"
_ICML.cc/2026/Conference — ICML 2026 regular_

### Official Review · Reviewer_xyW8 · 2026-03-08

**Soundness:** 4
**Presentation:** 3
**Significance:** 3
**Originality:** 2
**Overall Recommendation:** 5
**Confidence:** 4

**Summary:**

This paper studies out-of-distribution molecular representation learning and proposes MoSIR, which decomposes a molecular embedding into a prototype-based semantic component and an environment-dependent residual.

**Compliance With Llm Reviewing Policy:**

Affirmed.

**Final Justification:**

The authors fully resolved the issues raised in the review and thus I updated the overall evaluation from weak accept to accept. I am now fully happy to recommend acceptance of the paper.

**Key Questions For Authors:**

Do the gains survive a backbone swap, or is the method tied fairly tightly to the encoder used in the paper?

**Limitations:**

No. The paper should say more explicitly where the prototype bottleneck may hurt and what the practical cost of the min-max training is.

**Strengths And Weaknesses:**

The modeling makes sense and is aligned with the out-of-distribution molecular learning problem the paper discusses. The empirical evaluation is substantial, including both covariate and concept shifts, under different environment constructions and across both classification and regression tasks, while also comparing against a broad set of strong baselines, which makes the reported gains more convincing. The paper clarifies what the prototype space is capturing; the qualitative analyses of prototype selectivity, retrieved substructures, and physicochemical profiles help support the claim that the model is learning chemically meaningful structure rather than merely exploiting spurious correlations. Overall, the paper is well written and technically sound. The paper is not without limitations. Methods such as prototype-based projection, consistency regularization and adversarial perturbation are familiar, so the novelty lies more in the integration and molecular adaptation of them than in any fundamentally new method. Still, the work is technically solid and well motivated, providing significant experimental results, and thus I evaluate it positively towards acceptance.

---

> ### Author Rebuttal · Authors · 2026-03-30
>
> We sincerely thank the reviewer for the positive evaluation of our paper’s technical motivation, experimental design, and prototype-based interpretability analysis. In response to the reviewer’s concerns, we provide the following clarifications.
>
> **Strengths and Weaknesses.**
> - We agree with the reviewer’s opinion. that the novelty of this work does not mainly lie in proposing entirely new fundamental modules, but rather in organizing prototype projection, consistency regularization, and adversarial perturbation into a targeted and unified framework for the specific problem of **molecular OOD learning**.
> - More specifically, we explicitly decompose molecular representations into a stable semantic component and an environment-related residual component through a **prototype bottleneck**, thereby separating transferable information from environment-sensitive factors at the representation level. Furthermore, we apply perturbations only in the residual space, which more reasonably simulates environmental variations while avoiding direct destruction of the stable representation that serves as a semantic anchor.
> - In addition, we provide **theoretical analysis** showing how the prototype bottleneck improves OOD generalization by reducing the complexity of the hypothesis space. Combined with the consistent improvements on GOOD and DrugOOD under both covariate and concept shifts, as well as the prototype selectivity and substructure retrieval results, we believe that the main contribution of this work lies in effectively integrating these existing ideas in a way that is tailored to the molecular OOD problem, and in validating that the resulting model can indeed learn **chemically meaningful** and more robust representations.
>
> **Key Questions for Authors.**
> - MoSIR does not rely on a specific backbone encoder. Its core operations are semantic prototype decomposition and invariance learning performed on the **graph-level representations** produced by the encoder. To verify this, we replaced the default **GIN-Virtual** backbone with **GCN** on GOOD-ZINC, and further compared the performance gains over the baseline iMoLD. The results are shown in **Table 3**. As can be seen, MoSIR brings consistent improvements over iMoLD under both GIN-Virtual and GCN backbones, indicating that its benefit is not tied to any specific encoder.
>
> - Table 3. Backbone Generalization Analysis of MoSIR on GOOD-ZINC (MAE ↓, improvement over iMoLD in parentheses)
> | Split | GIN-Virtual | GCN | iMoLD |
> |---|---:|---:|---:|
> | scaffold / covariate | 0.0913 (+0.0497) | 0.0922 (+0.0488) | 0.1410 |
> | scaffold / concept | 0.0553 (+0.0461) | 0.0582 (+0.0432) | 0.1014 |
> | size / covariate | 0.1317 (+0.0546) | 0.1341 (+0.0522) | 0.1863 |
> | size / concept | 0.0637 (+0.0392) | 0.0664 (+0.0365) | 0.1029 |
>
> - In addition, **Section 4.4** of the paper applies MoSIR as a plug-in module to GTMGC (a Transformer backbone) for the ground-state conformation prediction task, and also achieves consistent improvements while keeping the backbone architecture and training settings unchanged. This further demonstrates that MoSIR has good applicability across different backbones.
>
> **Limitations:**
> - Regarding the potential negative effect of the prototype bottleneck, we agree that it may not be beneficial in all scenarios. If the task depends on **finer-grained 3D geometric or electronic structural information**, and such information cannot be sufficiently captured by shared semantic prototypes on 2D graphs, then the advantage of the prototype bottleneck may be weakened.
> - Regarding the practical computational cost, the main additional overhead comes from the inner maximization process in Eqs. (16)–(17), which is used to search for worst-case residual perturbations. This step is approximated by projected gradient ascent and therefore increases the training time. In contrast, the prototype projection itself is a relatively lightweight module. Importantly, this overhead mainly appears during training. At **inference time**, the model directly uses the learned semantic representation $z_G$ for downstream prediction without performing perturbation search, and thus does not introduce additional inference cost.

---

> > ### Author Rebuttal · Reviewer_xyW8 · 2026-03-31
> >
> > The authors fully addressed the issue raised in the review in great detail. It is fully resolved now.

---

> > > ### Author Response · Authors · 2026-04-01
> > >
> > > Dear Reviewer xyW8,
> > >
> > > We sincerely thank you for your positive feedback and for recognizing that we have fully addressed and resolved all of the concerns. We greatly appreciate the valuable time and careful effort you devoted to reviewing our manuscript, as well as your willingness to reconsider the score. Your comments have further refined our thinking and made the paper more solid.
> > >
> > > Sincerely,
> > > The Authors of Paper 13710

---

### Official Review · Reviewer_BWCe · 2026-03-12

**Soundness:** 3
**Presentation:** 3
**Significance:** 3
**Originality:** 3
**Overall Recommendation:** 5
**Confidence:** 4

**Summary:**

This paper proposes MoSIR, a novel framework for out-of-distribution generalization in molecular representation learning. The method decomposes molecular embeddings into a semantic invariant component and an environment-dependent residual. A bi-level min-max optimization is then employed to perturb the residual to simulate environment shifts. The approach is theoretically supported and empirically validated on the GOOD and DrugOOD benchmarks. Overall, the paper presents a solid contribution, though a few experimental details and analyses need clarification.

**Compliance With Llm Reviewing Policy:**

Affirmed.

**Final Justification:**

My concerns have been fully addressed, and the results are satisfactory.

**Key Questions For Authors:**

1. How sensitive is MoSIR to the choice of the underlying backbone? It would be interesting to see a brief discussion or ablation on how the framework performs when integrated with different architectures or, more importantly, with large-scale pre-trained molecular models. Do pre-trained backbones already capture sufficient semantic invariance, or does MoSIR still provide significant complementary gains?

2. Could the authors provide a brief analysis of why this specific parameter varies so significantly? Does a value as low as 0.002 imply that the Lintra constraint is less effective for certain tasks? Additionally, since this is an OOD setting, please briefly clarify the validation strategy used for hyperparameter tuning to ensure no information from the target OOD distribution was inadvertently utilized.

**Limitations:**

The authors have not discussed limitations or potential negative societal impacts. The paper would benefit from a brief discussion on the scope of MoSIR, for instance, which molecular types or datasets it is best suited for and where it might underperform.

**Strengths And Weaknesses:**

Strengths:
1. The concept of using a semantic prototype dictionary to create a representational bottleneck is intuitive and physically meaningful for molecular data.
2. The derivation of the OOD generalization bound (Theorem 3.1) and the complexity reduction via the prototype bottleneck (Proposition 3.2) provide a rigorous foundation for the empirical design.
3. The experimental section is comprehensive, demonstrating consistent improvements over strong baselines across multiple shift types (scaffold, size, assay).
Weaknesses:
1. While the method introduces a general GNN encoder, the specific backbone architecture used to obtain the main results (Tables 1 and 2) is not clearly specified in the main text.
2. Appendix Table 8 reveals substantial variation in hyperparameters across tasks. For example, the intra-molecular semantic consistency weight λ_intra is set to 0.3 for GOOD-HIV and GOOD-PCBA, but only 0.002 for GOOD-ZINC and DrugOOD datasets. This raises concerns about hyperparameter sensitivity and the underlying validation strategy.
3. Although the qualitative visualizations of top-activated molecules (Figures 3 and 4) are informative, the claim that the learned prototypes capture “invariant chemical semantics” would be more convincing if supported by quantitative evidence.
I recommend including a quantitative analysis, for instance, computing the average pairwise Tanimoto similarity (based on Morgan fingerprints) among molecules that strongly activate the same prototype. Such an evaluation would more rigorously demonstrate the structural consistency within the learned prototype space.

---

> ### Author Rebuttal · Authors · 2026-03-30
>
> We thank the reviewer for the valuable comments and constructive suggestions. We respond to the reviewer’s concerns below.
>
> **W1: GNN encoder.**
> - As stated in Section 4.1 (Experimental Setup) of our paper, following prior work **iMoLD**, we use **GIN-Virtual** as the backbone architecture for experiments on the GOOD dataset (Table 1), and **GIN** as the backbone architecture for experiments on the DrugOOD dataset (Table 2), in order to ensure a fair comparison with existing molecular OOD methods.
>
> **W2 and Q2: Hyperparameter sensitivity.**
> - In this paper, the coefficients $\lambda_{\text{intra}}$, $\lambda_{\text{inter}}$, and $\lambda_{\text{inv}}$ serve as loss balancing terms, and their effective scales depend on the task type and the numerical scale of the corresponding objectives. Therefore, a smaller $\lambda_{\text{intra}}$ does not mean that intra-molecular semantic consistency is ineffective; rather, it indicates that lighter reconstruction regularization is sufficient to stabilize the semantic bottleneck. Hence, these differences mainly reflect **optimization-scale** differences across different benchmarks, rather than indicating abnormal sensitivity of the method to hyperparameters.
> - In addition, the validation strategy adopted in our hyperparameter tuning is based on prior work **iMoLD**. The corresponding dataset splits are provided in **Table 5 of the appendix**. We strictly follow the benchmark train/validation/test split protocol, and do not use labels from the target OOD test set or any test-set statistical information.
>
> **W3: Quantitative analysis.**
> - On the GOOD-ZINC dataset under the size / concept split, we added a quantitative evaluation based on **Morgan fingerprints**. For each prototype $k$, we selected its highly activated molecule set $H_k$ (using top 5% / 10% / 20% activation thresholds, respectively), computed the average pairwise Tanimoto similarity among molecules within the set, and compared it with random molecule sets of the same size over 50 repetitions. We further report the corresponding bootstrap 95% confidence intervals. The results are shown in **Table 1**.
> - Table 1. Quantitative Structural Consistency of Learned Prototypes
> | High-Activation Rule | Intra-Prototype | Random Control | Δ (Intra - Random) | 95% CI of Δ |
> |---|---:|---:|---:|---:|
> | Top 5% activation  | 0.1296 | 0.1170 | +0.0126 | [0.0056, 0.0198] |
> | Top 10% activation | 0.1272 | 0.1170 | +0.0102 | [0.0031, 0.0173] |
> | Top 20% activation | 0.1242 | 0.1170 | +0.0072 | [0.0012, 0.0135] |
> - The results show that under all threshold settings, the intra-prototype similarity is consistently significantly higher than the random control, indicating that when focusing only on more strongly activated samples, the **internal structural consistency** of each prototype is further strengthened. This supports that the learned prototype space is not random clustering, but instead captures **chemically meaningful semantics** with structural consistency.
>
> **Q1: Backbone.**
> - In this paper, MoSIR only requires graph-level representations as input, and does not depend on any specific backbone network. To verify this, we replaced GIN-Virtual with GCN on the GOOD-ZINC dataset. The experimental results, shown in **Table 2**, indicate that our method maintains stable performance across different GNN backbones.
> - Table 2. Backbone Sensitivity Analysis on GOOD-ZINC (MAE ↓)
> | Split | GIN-Virtual | GCN |
> |---|---:|---:|
> | scaffold / covariate | 0.0913 | 0.0922 |
> | scaffold / concept | 0.0553 | 0.0582 |
> | size / covariate | 0.1317 | 0.1341 |
> | size / concept | 0.0637 | 0.0664 |
> - In addition, as shown in Section 4.4 (ground-state conformation prediction) of the paper, we have already applied MoSIR as a plug-in module to **GTMGC**(a Transformer architecture). The experimental results are reported in **Table 7 of the appendix**, and they show that our method achieves consistent performance improvements.
> - Regarding pretrained backbones, we agree that stronger pretrained models may already capture part of the stable semantics. However, their representations usually do not explicitly distinguish semantic invariant factors from environment-sensitive variations, which is exactly the core aspect modeled by MoSIR. Therefore, we believe that MoSIR is complementary to pretrained molecular models: the former provides a stronger representational foundation, while the latter further enhances OOD generalization.
>
> **Limitations:**
> - We believe that MoSIR is more suitable for molecular tasks where transferable semantic factors remain predictive under scaffold, size, or assay shifts. Its advantage may weaken when the target property depends heavily on **3D conformations or electronic effects**. Improving robustness under distribution shift may benefit molecular screening and scientific discovery, but MoSIR predictions should be regarded as computational support rather than a substitute for experimental validation.

---

> > ### Author Rebuttal · Reviewer_BWCe · 2026-04-03
> >
> > The authors have clearly addressed all my concerns, and given the solid results.

---

> > > ### Author Response · Authors · 2026-04-03
> > >
> > > Dear Reviewer BWCe,
> > >
> > > Thank you for your time and careful review. We are very pleased that you recognized our response. Your suggestions have been very important in improving the paper.
> > >
> > > Sincerely,
> > >
> > > The Authors of Paper 13710

---

### Official Review · Reviewer_fgtM · 2026-03-13

**Soundness:** 3
**Presentation:** 3
**Significance:** 2
**Originality:** 3
**Overall Recommendation:** 4
**Confidence:** 2

**Summary:**

Molecular OOD learning remains challenging as training data typically covers only a limited portion of the chemical space. MoSIR is proposed for learning semantic invariant representation, and a bi-level min-max objective is introduced to simulate plausible environment shifts and enforce semantic stability.

Experiments on GOOD benchmarks demonstrate that MoSIR outperforms baselines across shift settings.

**Compliance With Llm Reviewing Policy:**

Affirmed.

**Final Justification:**

From the second round of the authors' response, the following concerns are better resolved:
1. The OOD scaffold test, which is shown in Table 1.
2. The prototype number $K$ has been tested through an ablation study;

However, the following key concerns are still there:
1. The key motivation that the larger datasets have limited chemical space is still not solid. The distribution shift (mainly based on the labels) itself can be regarded as OOD; however, it is totally different from the chemical space aspect, such as the scaffold or other chemical features. So, overall, the problem definition is weak, undermining the foundation of the whole paper.
2. Although the author added quantitative analysis beyond Figure 2&3, it is still not going straight into the insights of the key novelty: prototypes. The reason for the prototype design is neither intuitively clear nor experimentally validated.

Overall, I think this paper is generally not bad for acceptance, but it is far from a rewarding paper, such as an oral or spotlight. I keep the weak positive score.

**Key Questions For Authors:**

1.	What's the definition of OOD in this work? Does a different scaffold count as an OOD sample?
2.	Can you show more insights on how the model intuitively captures semantic invariance under consistency constraints?

**Limitations:**

The limitations of this paper mainly focus on the motivation, for instance, how to prove that the challenge that limited chemical space causes models to rely on environment-dependent factors. It's better to prove or illustrate the existing challenges more clearly.

**Strengths And Weaknesses:**

Strengths:

1.	Experiments on GOOD benchmarks are plausible to demonstrate the advantage of OOD performance of MoSIR. Experiments are clearly defined in the header of Sec. 4 and are answered logically in the following subsections.
2.	Ablations in Table 3 clearly indicate that the proposed two novelties in MoSIR have positive gains.
3.	Extensive theoretical analysis in Sec. 3.6 and Appendix A are rigorous and well-organized.
4.	Overall, the presentation of this paper is clean and easy to follow. One small suggestion is that Figure 1 should be more self-explanatory. Currently, it's hard to read the main idea from Fig. 1 before reading the whole definition and equations in Sec. 3.
5.	MoSIR solves the functional compositions shift problem by enforcing semantic invariance and mitigating spurious correlations, which is straight to the problem and in the right direction.

Weaknesses:

1.	This paper is built on the motivation that training data only covers a limited portion of the chemical space, as claimed in the abstract. However, large molecule databases, such as Zinc, can have diverse scaffold molecules for self-supervised training.
2.	In equation 19, the OOD generation bound. The Q is the arbitrary target distribution that is usually hard to estimate in real-world cases. How does the unknown Q influence the bound?
3.	There are k prototypes in the Semantic Prototype Decomposition. There is no evidence showing how large the k should be to cover an acceptable chemical space.

---

> ### Author Rebuttal · Authors · 2026-03-30
>
> We thank the reviewer for the helpful comments. Regarding Figure 1, we will revise its structure and caption to make the main idea of our method clearer. Our detailed responses to the remaining comments are provided below.
>
> **W1: Motivation.**
> - We agree that large-scale molecular databases such as ZINC exhibit substantial molecular diversity. However, our motivation is not about whether the dataset size is sufficiently large, but rather about **its limited coverage of the true chemical space**. Even large datasets only cover a small subset of the chemical space, where certain scaffolds, substructures, and functional group combinations appear frequently, while many plausible chemical structures are rarely observed or even missing.
> - As a result, models may rely on dominant chemical patterns in the training distribution and fail to generalize to unseen scaffolds or compositional variations. Our work is designed to address this issue by explicitly separating stable **semantic factors** from environment-sensitive variations, thereby improving generalization under unseen distributions.
>
> **W2: OOD generation bound.**
> - We agree that in real-world scenarios, the target distribution $Q$ is typically unknown. However, it is important to emphasize that our generalization bound follows standard domain generalization theory and does not rely on explicitly estimating $Q$. The role of $Q$ is to characterize the **risk gap** between the training distribution and potential test distributions.
> - Our method reduces the **hypothesis complexity** via the prototype bottleneck and mitigates distribution shift through semantic invariance constraints under residual perturbations. Therefore, even when $Q$ is unknown, the bound still provides a theoretical explanation for the improved OOD generalization of MoSIR.
>
> **W3: Semantic prototype.**
> - In fact, the role of $K$ is not to exhaustively cover the entire chemical space, but to control the **capacity of the semantic bottleneck**.
> - We conducted a sensitivity analysis on GOOD-ZINC with $K$ $\in$ \{5, 10, 20, 30, 40\}. The results show that when $K$ is too small, the prototype dictionary cannot capture diverse chemical semantics, while increasing $K$ steadily improves performance until it reaches the best result at a moderate size ($K$). When $K$ becomes too large, redundant prototypes are introduced, which leads to unstable optimization and slight performance degradation.
> - Therefore, $K$ should be understood as a **capacity hyperparameter** of the semantic bottleneck rather than a measure of chemical space coverage.
>
> **Q1: The definition of OOD.**
> - In this paper, OOD refers to the **distribution shift** between training and test environments.
> - In benchmarks such as GOOD and DrugOOD, OOD is not defined by checking whether individual samples satisfy certain shift conditions, but is constructed through predefined data splits (e.g., scaffold split, size split) that induce distribution differences between training and test sets.
> - Therefore, under the **scaffold-based setting**, molecules in the test set with scaffolds different from those in the training set are regarded as OOD samples.
>
> **Q2: Semantic invariance.**
> - MoSIR decomposes molecular representations into a semantic component $z_G$ and a residual component $e_G$, where $z_G$ captures stable semantics and $e_G$ encodes environment-related variations. We apply perturbations only in the residual space and enforce consistency on $z_G$ before and after perturbation, encouraging the model to encode **generalizable semantic information** into $z_G$.
> - Section 4.3 provides intuitive evidence, Figure 2 shows that learned prototypes align well with typical functional groups, and Figure 3 demonstrates that molecules strongly activating the same prototype share consistent substructure patterns. These results indicate that the consistency constraint helps the model capture **stable chemical semantics**.
>
> **Limitations:**
> - Our motivation is that training data typically cover only a limited portion of the true chemical space. Under such conditions, certain scaffolds, substructures, or functional group combinations may form dominant predictive cues in the training environment, causing models to rely on environment-sensitive factors and struggle to generalize to unseen distributions.
> - Benchmarks such as **GOOD**  and **DrugOOD**  have shown that GNNs with strong in-distribution performance degrade significantly under scaffold or other environment-based splits.
> - Consistently, our results in Tables 1 and 2 also show that standard ERM performs poorly under various OOD settings. These observations jointly support our motivation.

---

> > ### Author Rebuttal · Reviewer_fgtM · 2026-04-03
> >
> > For weakness 1, I know that this work assumes even for a larger dataset, the coverage of the chemical space is not enoughl. However, there is no evidence showing the claim is true. Many of the chemical rules are extracted based on the large molecule databases. And the databases are nearly include all the molecules that discovered by humans. So, if you say the coverage is not enough, solid evidence is needed.
> >
> > For the Q in generation bound, the authors' explanation may make sense.
> >
> > For the hyperparameter K in the semantic prototype, is the best-k different in different tasks? How the melecular size influences the best-k?
> >
> > For Q1, did you test the OOD performance for different scaffolds? For example, a property dataset that is splitted by scaffold.
> >
> > For Q2, to be honest, the question is right about the Figure 2. It's quite hard for readers to see any insight from it about the semantic alignment. The same thing can be found in Figure 3, how can we see the demostrated molecules are showing the same functional group pattern? The structures still look different for all of them. How to define and quantify the consistency of substructures?
> >
> > So actually most of the concerns are still there, my positive assessment mainly comes from the paper's illustration skills, positive experiment results, and the motivation.

---

> > > ### Author Response · Authors · 2026-04-04
> > >
> > > We sincerely thank you for your positive comments on the manuscript’s illustration skills, positive experimental results, and motivation, as well as for your detailed follow-up feedback. We respond to your concerns as follows.
> > >
> > > **W1:**
> > > - As you noted, large-scale molecular datasets contain abundant scaffold molecules and have been widely used for self-supervised pretraining; therefore, from this perspective, the training data provides fairly broad coverage of the chemical space. **However, this paper focuses on the OOD generalization problem in molecular representation learning**, namely, how models can learn more stable representations when the training and test distributions differ. Therefore, consistent with existing OOD molecular learning works [1,2], the benchmark we adopt simulates the situation where the training data has limited coverage of task-relevant structural patterns, which is also why models can easily rely on environment-related factors and degrade under unseen distributions.
> > >
> > > - In addition, even if large-scale pretrained models may have captured some stable semantics, their representations usually do not explicitly distinguish semantic-invariant factors from environment-sensitive variations, **which is exactly the core issue modeled by MoSIR**. In the abstract of the original paper, we stated that the training data usually covers only a limited part of the chemical space; this statement is intended for the OOD setting, and we will clarify this more explicitly in the revised version.
> > >
> > > [1] Zhuang et al., iMoLD, NeurIPS 2023.
> > >
> > > [2] Wu et al., CFD, ICLR 2025.
> > >
> > > **W3:**
> > > - **Optimal $K$ values across different tasks.** Appendix Table 8 in the original paper shows that the optimal $K$ values vary across tasks and datasets, but generally fall around $K=20$ or $K=30$. Furthermore, our theoretical analysis shows that the model complexity grows only with $\log K$, which theoretically explains why a moderate-scale $K$ is sufficient to achieve stable performance across different tasks.
> > >
> > > - **How molecular size affects the optimal $K$.** We further conduct supplementary verification under the GOOD-HIV size/covariate setting. Specifically, we define molecular size as the number of atoms in a molecule (node size), and divide the molecules into three subgroups, Small/Medium/Large, according to the training-set quantiles. Keeping all other training configurations fixed, we perform a grid search over $K$ $\in$ \{5,10,20,30,40\}, and then use OOD validation to select the optimal $K$ for each size subgroup. The results are shown in Table 1 below, where the optimal $K$ exhibits a clear increasing trend with molecular size, while remaining within a moderate range overall, and larger $K$ does not lead to continued performance gains.
> > >
> > > - Table 1. Sensitivity of ROC-AUC (%) to Prototype Number $K$ across Molecular Size Subgroups
> > > |$K$|Small|Medium|Large|
> > > |---:|---:|---:|---:|
> > > |5|75.64|76.43|77.08|
> > > |10|**76.23**|76.91|77.54|
> > > |20|75.97|**77.16**|77.93|
> > > |30|75.73|76.95|**78.12**|
> > > |40|75.95|76.84|77.86|
> > >
> > > **Q1:**
> > > - Yes, **Table 1 in the original paper** has already explicitly evaluated the performance under scaffold-based OOD settings. Specifically, GOOD-HIV, GOOD-ZINC, and GOOD-PCBA all include two scaffold-based OOD settings: scaffold/covariate and scaffold/concept.
> > >
> > > **Q2:**
> > > - **Figure 2** in the original paper shows stable statistical associations between prototypes and typical chemical semantics. For example, P15 is positively selective for Amide, N, and NH, but negatively selective for Thiol, suggesting an association with nitrogen-containing and amide/hydrogen-bond-related semantics. **Figure 3** further supports this at the molecular level: top-activated molecules of the same prototype often share similar local structural patterns. For P15, these molecules typically contain aromatic/heteroaromatic cores with linking groups such as C=O and N, consistent with the pattern shown in Figure 2.
> > >
> > > - We agree that Figures 2 and 3 are mainly qualitative. We therefore add a **quantitative substructure consistency analysis** on GOOD-ZINC size/concept using Morgan fingerprints. For each prototype $K$, we select its top 5%/10%/20% activated molecules, compute the average pairwise Tanimoto similarity, and compare it with random sets of the same size over 50 runs, with bootstrap 95% CIs.
> > >
> > > - Table 2. Quantitative Structural Consistency of Learned Prototypes
> > > |High-Activation Rule|Intra-Prototype|Random Control|Δ (Intra - Random)|95% CI of Δ|
> > > |---|---:|---:|---:|---:|
> > > |Top 5% activation|0.1296|0.1170|+0.0126|[0.0056, 0.0198]|
> > > |Top 10% activation|0.1272|0.1170|+0.0102|[0.0031, 0.0173]|
> > > |Top 20% activation|0.1242|0.1170|+0.0072|[0.0012, 0.0135]|
> > >
> > > The results of Table 2 show higher within-prototype similarity under all thresholds, especially stricter ones, supporting that the learned prototype space captures structurally consistent chemical semantics rather than random clusters.

---

### Decision · Program_Chairs · 2026-04-30

**Decision:**

Accept (regular)

**Comment:**

This paper presents MoSIR, a prototype-based framework for molecular OOD generalization that decomposes embeddings into a semantic-invariant component and an environment-dependent residual, trained via a bi-level min-max objective with adversarial perturbations restricted to the residual space. All three reviewers acknowledge the solid empirical gains on GOOD and DrugOOD across covariate and concept shifts (Reviewers fgtM, BWCe, xyW8), the theoretical analysis grounding the prototype bottleneck (Reviewers fgtM, BWCe), and the interpretability of the learned prototypes (Reviewer xyW8). The main concerns covered the motivation around limited chemical-space coverage (Reviewers fgtM), backbone sensitivity and hyperparameter variance across datasets (Reviewers BWCe, xyW8), and the qualitative nature of the prototype visualizations (Reviewers fgtM, BWCe). The rebuttal added Tanimoto-based quantitative consistency analysis, a GCN backbone comparison, a plug-in result on GTMGC, and a prototype-number sensitivity study. Reviewers fgtM remains partially skeptical about the motivational framing but retained a weak accept. The remaining concern is soft and does not undermine the empirical contribution. I support acceptance.